# Molecular basis for dyneinopathies reveals insight into dynein regulation and dysfunction

Matthew G Marzo[1], Jacqueline M Griswold[1], Kristina M Ruff[1], Rachel E Buchmeier[1], Colby P Fees[2], Steven M Markus[1]*

[1]Department of Biochemistry and Molecular Biology, Colorado State University, Fort Collins, United States; [2]Department of Cell and Developmental Biology, University of Colorado School of Medicine, Aurora, United States

**Abstract** Cytoplasmic dynein plays critical roles within the developing and mature nervous systems, including effecting nuclear migration, and retrograde transport of various cargos. Unsurprisingly, mutations in dynein are causative of various developmental neuropathies and motor neuron diseases. These 'dyneinopathies' define a broad spectrum of diseases with no known correlation between mutation identity and disease state. To circumvent complications associated with dynein studies in human cells, we employed budding yeast as a screening platform to characterize the motility properties of seventeen disease-correlated dynein mutants. Using this system, we determined the molecular basis for several classes of etiologically related diseases. Moreover, by engineering compensatory mutations, we alleviated the mutant phenotypes in two of these cases, one of which we confirmed with recombinant human dynein. In addition to revealing molecular insight into dynein regulation, our data provide additional evidence that the type of disease may in fact be dictated by the degree of dynein dysfunction.
DOI: https://doi.org/10.7554/eLife.47246.001

*For correspondence:
steven.markus@colostate.edu

Competing interests: The authors declare that no competing interests exist.

## Introduction

Motor-mediated intracellular transport is essential for numerous critical cellular processes (*Roberts et al., 2013*; *Kardon and Vale, 2009*; *Vallee et al., 2012*). This is especially apparent in motor neurons, in which cargoes must be transported over long distances to support various neuronal functions. For instance, the soma is the primary site of metabolic function where RNAs and proteins are synthesized and degraded. Thus, to maintain neuronal health, it is critical that cargoes are transported from the soma to the axon terminus (*i.e.*, along the axon) and vice versa. For example, neurofilaments, which provide structural stability to a cell, are transported to the axon terminus by plus end-directed microtubule motors (kinesins; *Xia et al., 2003*) and to the soma by the minus end-directed microtubule motor, cytoplasmic dynein (hereafter referred to as dynein) (*He et al., 2005*; *Wagner et al., 2004*; *Shah et al., 2000*). Dynein has also been shown to play a key role in trafficking numerous other neuronal cargoes in various model organisms (*He et al., 2005*; *Wagner et al., 2004*; *Shah et al., 2000*; *Bowman et al., 2000*; *Berg et al., 2000*; *Barkus et al., 2008*; *Hendricks et al., 2010*; *Rosa-Ferreira and Munro, 2011*; *Maday et al., 2012*; *Kamal et al., 2000*; *Lazarov et al., 2005*; *Fu and Holzbaur, 2013*; *Almenar-Queralt et al., 2014*; *Rao et al., 2017*), including autophagosomes (*Ravikumar et al., 2005*; *Katsumata et al., 2010*; *Cheng et al., 2015*), mitochondria, and ionotropic glutamate receptors (*Horak et al., 2014*). Defects or perturbations in dynein function lead to mislocalization of these cargoes, and results in an accumulation of protein aggregates at the neurite tip (*Ravikumar et al., 2005*; *Levy et al., 2006*). In fact, accumulation of misfolded protein aggregates in the neuronal cytoplasm is a common pathological hallmark for

**eLife digest** Motor proteins maintain order by transporting biomolecules and various structures within living cells. Dynein is one such motor that moves many types of cargoes along tracks called microtubules, which are spread across the cell's interior. This motor is particularly important in nerve cells, which can be very long and thus depend heavily on motor proteins to ensure cargoes end up where they are needed. This becomes especially apparent in human diseases that arise as a consequence of mutations in the genes that produce components of the dynein motor.

It is assumed that these genetic changes simply prevent dynein from working properly, which ultimately affects the health and survival of cells. However, it is currently unknown what specific effect these mutations have on dynein's role within the cell, and how these changes lead to particular diseases.

Marzo et al. have now used dynein from a budding yeast to closely examine 17 mutations in the dynein gene that are associated with developmental and/or motor neuron diseases in humans. For each mutation, various aspects of how dynein moves (e.g. average speed, distance travelled) were measured and quantitatively compared. The results show that the severity of the effect of each mutation can be directly correlated with the type of disease caused by the mutation. In particular, mutations that lead to less severe defects are found in patients that suffer from various motor neuron diseases, while more severe dynein mutations are found in patients with developmental brain disorders. Marzo et al. confirmed the likely structural changes that caused the defects in dynein's activity in two of the 17 cases, by engineering additional, restorative mutations that lessened the effects of the primary mutation.

These findings reveal links between the molecular impact of defects in the dynein gene and human health. They also confirm that budding yeast is a powerful tool for investigating newly discovered dynein mutations that correlate with disease. This study provides a potential system that could be used to screen drugs that might lessen the effects of specific dynein mutations. However, further work is needed to determine how effective this system will be for drug discovery.

DOI: https://doi.org/10.7554/eLife.47246.002

motor neuron disease (*He et al., 2005*; *LaMonte et al., 2002*; *Lin and Schlaepfer, 2006*). Consistent with an important role for dynein in neuronal health (*Schiavo et al., 2013*), mouse models with dynein mutations exhibit severe neuropathy, and decreased rates of retrograde axonal transport, among other defects (*Hafezparast et al., 2003*; *Chen et al., 2007*; *Ori-McKenney et al., 2010*).

In addition to its key role in the retrograde trafficking of cargoes, dynein also plays a critical and conserved role during neuronal development by promoting interkinetic nuclear migration (INM) in neuronal progenitor cells, and in the subsequent migration of young postmitotic neurons. During the former process, which is critical for neurogenesis, nuclear envelope anchored dynein motors promote migration of the nucleus from the basal to the apical surface of the neuroepithelium where mitotic divisions occur (*Tsai et al., 2010*; *Hu et al., 2013*; *Del Bene et al., 2008*). Thus, defects in dynein-mediated nuclear migration can lead to defects in early brain development.

Given its myriad roles in neuronal processes, it is unsurprising that mutations within the catalytic component of the dynein complex (the dynein heavy chain, or DHC, which is encoded by the DYNC1H1 gene) are found in individuals suffering from a wide array of neurodegenerative diseases. For instance, mutations in dynein underlie many cases of malformations of cortical development (MCD), spinal muscular atrophy with lower extremity dominance (SMA-LED), congenital muscular dystrophy (CMD), and Charcot-Marie-Tooth disease (CMT) (*Harms et al., 2012*; *Tsurusaki et al., 2012*; *Scoto et al., 2015*; *Peeters et al., 2015*; *Strickland et al., 2015*; *Weedon et al., 2011*; *Fiorillo et al., 2014*). Although mutations in the dynein regulator LIS1 are causative of the MCD disease lissencephaly (*Wynshaw-Boris, 2007*; *Wynshaw-Boris and Gambello, 2001*; *Reiner et al., 1993*) (characterized by a smooth brain due to reduced or absent cortical folding), mutations in dynein more often lead to polymicrogyria, which is characterized by excessive small folds in the cerebral cortex (*Poirier et al., 2013*; *Laquerriere et al., 2017*; *Willemsen et al., 2012*; *Vissers et al., 2010*). Although a clear link has been established between dynein dysfunction and various neurological diseases, the underlying molecular basis for disease onset or progression is unknown.

An unambiguous mechanistic dissection of mutant dynein function in animal cells is complicated by the diverse cellular functions in which dynein participates (*e.g.*, axonal transport, centrosome separation, spindle assembly, nuclear envelope breakdown, spindle checkpoint inactivation) (*Rusan et al., 2002*; *Gönczy et al., 1999*; *Salina et al., 2002*; *Howell et al., 2001*; *Wojcik et al., 2001*; *Chevalier-Larsen and Holzbaur, 2006*), and the difficulties and expense associated with generating and analyzing mutant cell lines (*e.g.*, compromised viability, pleiotropism, heterozygosity). To overcome these issues, we have employed the versatility and power of the budding yeast *Saccharomyces cerevisiae* to understand how mutations found in individuals suffering from various neurological diseases lead to dynein dysfunction. In addition to their genetic amenability, low maintenance costs, and rapid generation time, the study of dynein function in budding yeast is simplified by several factors. In contrast to animal cells in which dynein performs numerous functions, the only known function for dynein in budding yeast is to position the mitotic spindle at the future site of cytokinesis (*Li et al., 1993*; *Eshel et al., 1993*; *Carminati and Stearns, 1997*), making functional studies of dynein mutants in this organism simple and unambiguous. As in higher eukaryotes, the yeast dynein complex is comprised of light (Dyn2), light-intermediate (Dyn3), intermediate (Pac11), and heavy chains (Dyn1), the latter of which is the ATPase that powers motility along microtubules (see *Figure 1A*) (*Markus and Lee, 2011a*). Whereas in humans, the non-catalytic subunits exist in different isoforms encoded by multiple genes and tissue-specific isoforms (*Pfister et al., 2006*; *Raaijmakers et al., 2013*), each of the accessory chains in budding yeast is encoded by only a single gene, enabling simple genetic analysis and manipulation. Moreover, studies have revealed a high degree of structural similarity between yeast and human dynein (*Carter, 2013*; *Schmidt and Carter, 2016*), rendering structure-function studies in this organism relevant and translatable to animal cells. Compounded by the genetic amenability, ease of imaging, and the simple one-step method for isolation of recombinant, motile dynein motors (*Reck-Peterson et al., 2006*; *Markus et al., 2012*; *Markus and Lee, 2011b*), budding yeast are a powerful model system for studies of dynein function.

For this study, we focused on a library of 17 single point mutations in the dynein heavy chain (Dyn1), which are found in patients suffering from a broad spectrum of neurological diseases (*Figure 1A*). These mutations were selected based on their conservation with corresponding residues in yeast dynein. Our findings reveal phenotypic signatures associated with the mutant library that range from partial to complete loss-of-function, and even gain-of-function in some respects. In some cases, the altered function was intrinsic to dynein, while in others, the effects could be attributed to alterations in the activity of the holo-dynein-dynactin complex. The rapid nature of our phenotypic analysis combined with the wealth of structural information available for dynein has enabled us to assess the likely structural basis for dysfunction in two instances. Moreover, consistent with recent findings from another group (*Hoang et al., 2017*), our work reveals a correlation between the degree of dynein dysfunction and disease type. Overall, we describe a rapid and unambiguous set of tools that can be used to understand how disease-correlated mutations in dynein genes lead to onset or progression of disease.

## Results

### Budding yeast as a model organism for dynein dysfunction

Genomic alterations in individuals suffering from a variety of neurological diseases have been mapped to numerous unique sites throughout the dynein heavy chain (*Figure 1A*). The tail domain (~1400 amino acids) is the site of interaction for accessory chains (light-intermediate and intermediate chains), as well as adaptors that link dynein to the dynactin regulatory complex and various cellular cargoes (*Urnavicius et al., 2015*). The motor domain (~3000 amino acids) forms the catalytic core of dynein, where ATP binding and hydrolysis is translated into movement of the linker element that powers motility along microtubules (*Carter, 2013*; *Schmidt et al., 2015*; *Roberts et al., 2013*). Although the genetic basis for dyneinopathies is known, there exists very limited data on how these mutations are causative of motor dysfunction. Indeed, the mutations map throughout the entire heavy chain, with no clear correlation between disease state (*i.e.*, symptoms, severity, age of onset) and mutation identity. To understand how disease-correlated point mutations affect dynein function,

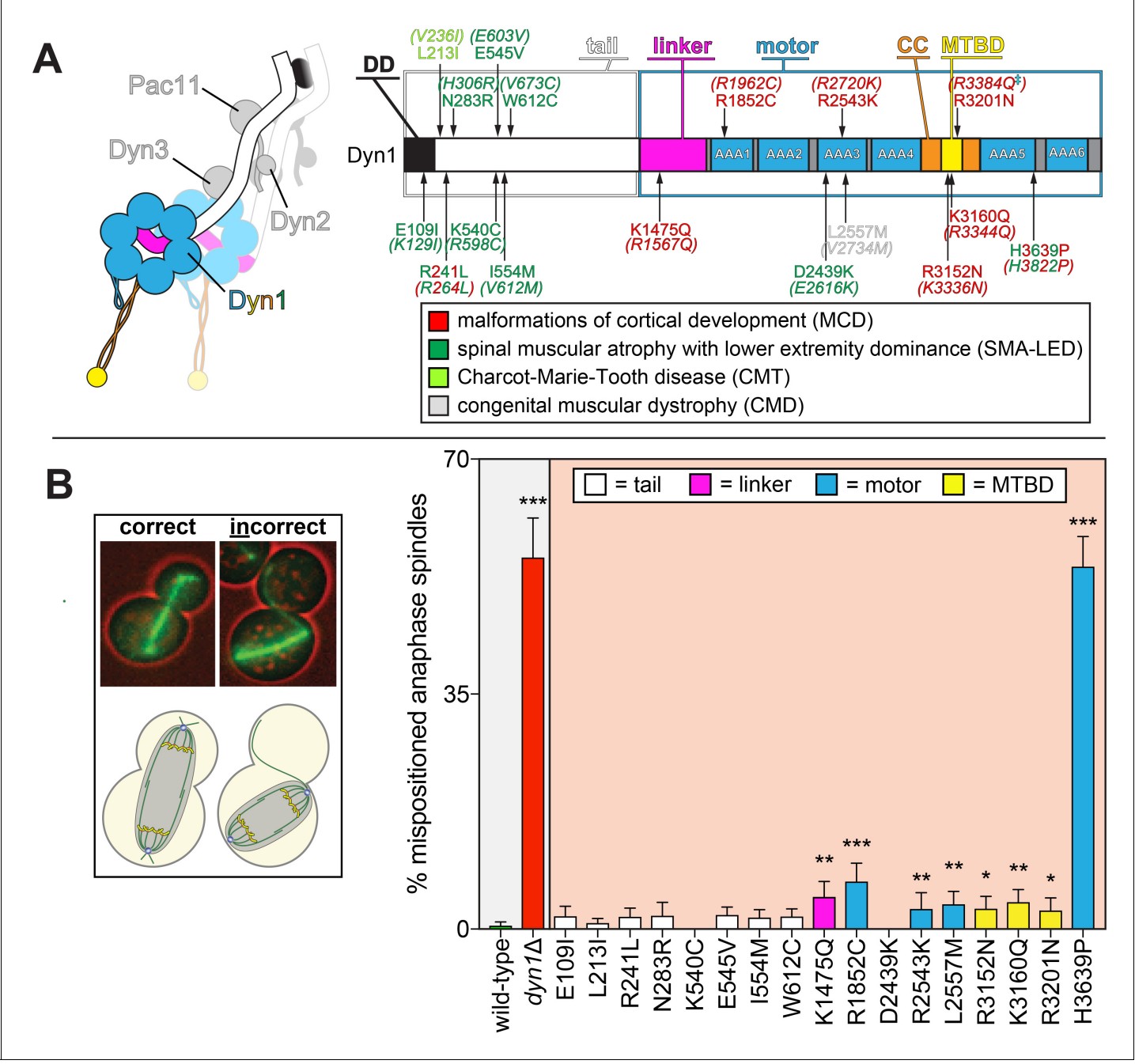

**Figure 1.** Spindle positioning assay provides coarse assessment of mutant dynein dysfunction. (**A**) Color-coded cartoon representation of the full-length dynein complex (left; with associated accessory chains; Dyn2, dynein light chain; Dyn3, dynein light-intermediate chain; Pac11, dynein intermediate chain; Dyn1, dynein heavy chain), and a linear schematic of Dyn1 with indicated disease-correlated mutations (right; DD, dimerization domain; CC, coiled-coil; MTBD, microtubule-binding domain). The equivalent human residues and disease-correlated substitutions are indicated in parentheses for each residue. ‡Note that we mistakenly substituted an asparagine for residue R3201 instead of a glutamine, the latter of which was identified as correlating with MCD (***Poirier et al., 2013***). R3201N was used throughout this study. (**B**) Representative fluorescence image (left; green, GFP-Tub1; red, contrast enhanced brightfield image to illustrate cell cortex) and quantitation of spindle positioning phenotypes in the 17 disease-correlated Dyn1 mutants, along with wild-type and dynein knock-out (*dyn1Δ*) cells. Each data point represents the fraction of mispositioned anaphase spindles along with standard error (weighted mean ± weighted standard error of proportion; n ≥ 99 anaphase spindles from three independent experiments for each strain). Statistical significance was determined by calculating Z scores, as described in the Materials and methods (*, p≤0.1; **, p≤0.05; ***, p≤0.001). Also see ***Figure 1—source data 1***.

DOI: https://doi.org/10.7554/eLife.47246.003

*Figure 1 continued on next page*

*Figure 1 continued*

The following source data is available for figure 1:

**Source data 1.** Spreadsheet with spindle positioning assay values.

DOI: https://doi.org/10.7554/eLife.47246.004

we employed a series of well-established methodologies to assess the function of 17 single point mutants in budding yeast.

The first such assessment was to determine if the mutant motors were capable of correctly positioning the mitotic spindle, the only known function for dynein in budding yeast. To this end, we performed a spindle positioning assay using haploid yeast cells expressing mRuby2-Tub1 (α-tubulin; to visualize the mitotic spindle) and the mutant dynein from the native dynein locus. Given the dispensable nature of dynein function for yeast cell viability (*Li et al., 1993*; *Eshel et al., 1993*), a complete loss-of-function mutant would not be expected to compromise viability, but to simply affect spindle position. In this assay, single time-point images of mutant cells are acquired, and the position of the mitotic spindle is deemed to be either correct (*i.e.*, the anaphase spindle extended through the bud neck along the mother-bud axis) or incorrect (see *Figure 1B*, left).

This analysis revealed that eight mutants exhibited varying degrees of spindle positioning defects that significantly differed from wild-type (*Figure 1B*, right). All of those mutants with defects were those with substitutions in the motor domain, which encompasses the six AAA (ATPase associated with various cellular activities) domains, the linker element that performs the powerstroke, and the microtubule-binding domain (MTBD). All but one of the motor domain mutations led to significant defects. Seven out of the nine mutations linked to malformations in cortical development (MCD) were among those with defects in this assay, whereas none of the mutations associated with the other neurological diseases exhibited significant defects. Although most of these mutants differed to a small but significant degree from wild-type, H3639P exhibited defects as severe as loss of *DYN1* (*dyn1Δ*).

## Spindle tracking in live cells provides a sensitive readout for dynein-dynactin dysfunction

Given the somewhat binary nature of the spindle positioning assay, it provides only a coarse assessment of mutant functionality. Thus, as a more sensitive readout of mutant dynein function, we imaged dynein-mediated spindle movements in yeast cells and quantitatively assessed various parameters of these movements. Given the reliance of dynein on dynactin for this activity in cells, assessment of spindle movements is in fact a read-out of dynein-dynactin activity. To ensure that spindle translocation events were a consequence of dynein-dynactin activity, we performed these assays in cells deleted for *KAR9*, a genetic component of a pathway that promotes orientation of the spindle along the mother-bud axis (*Yin et al., 2000*; *Hwang et al., 2003*; *Liakopoulos et al., 2003*) (see *Figure 2—figure supplement 1*). Moreover, we treated these cells with hydroxyurea (HU), an inhibitor of DNA synthesis that arrests yeast in a prometaphase-like state that precludes anaphase onset. This relatively simple but sensitive assay permits detection of subtle defects (or enhancements) in the motility parameters of dynein-dynactin (*Moore et al., 2009*). In addition to obtaining velocity and displacement values, this assay also provides a readout for relative activity (*i.e.*, how active dynein-dynactin is within the cell). Moreover, by scoring for 'neck transit' success frequency – events in which a dynein-dynactin-mediated spindle translocation results in successfully transiting the narrow mother-bud neck – we are also able to determine if there are potential defects in force production. Previous studies have shown that neck transits are compromised in cases where dynein's microtubule-binding affinity is weakened (*Ecklund et al., 2017*), or when the CAP-gly domain of Nip100 (homolog of human p150 component of the dynactin complex) is genetically ablated (*Moore et al., 2009*). Subsequent to HU arrest, full Z-stacks of the mitotic spindle and astral microtubules (via GFP-Tub1) were acquired every 10 s (see *Figure 2A* for example), and the position of the spindle was subsequently tracked using a combination of manual (*e.g.*, to assess neck transit success) and automated 3-dimensional tracking (using a custom written Matlab-based routine).

This analysis revealed a broader more nuanced array of defects in our library of mutants (*Figure 2B–F*, and *Figure 2—figure supplement 2A and B*). Strikingly, all mutants exhibited varying

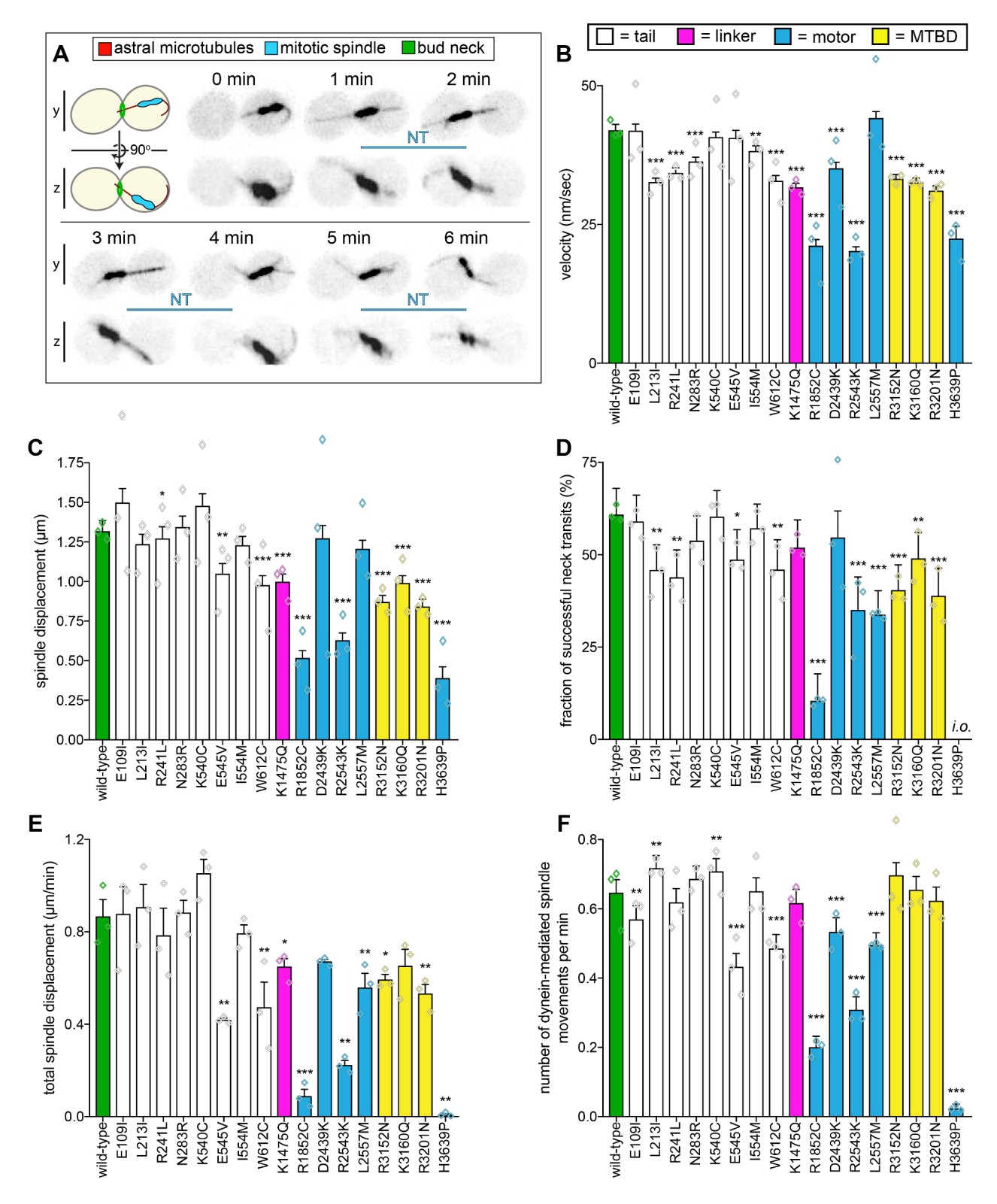

**Figure 2.** Quantitative assessment of dynein-dynactin-mediated spindle dynamics reveals refined insight into mutant dysfunction. (**A**) Cartoon and representative time-lapse inverse fluorescence images of a hydroxyurea (HU)-arrested *kar9Δ* cell exhibiting typical dynein-mediated spindle movements, analysis of which is presented in panels (**B – F**). Maximum intensity (X-Y projection; top) and Y-Z projections (bottom) are shown for each time point (NT, neck transit; note, line spans time frames over which the NT occurs). (**B – F**) Plots of indicated parameters for spindle dynamics in *Figure 2 continued on next page*

*Figure 2 continued*

haploid wild-type and mutant strains. Briefly, the mitotic spindles were tracked in 3-dimensions using a custom written Matlab code. Dynein-mediated spindle movements were manually selected from the tracking data, from which velocity (B), displacement (C, per event; or, E, per minute), and the number of dynein-mediated spindle movements per minute (F) were obtained. The fraction of successful neck transits (successful attempts divided by total attempts) were manually scored (*i.o.*, insufficient observations; for H3639P, only one unsuccessful neck transit attempt was observed). Each bar represents the weighted mean ± weighted standard error (or standard error of proportion for D; n = 42 to 60 HU-arrested cells from three independent experiments were analyzed for each strain; diamonds represent mean values obtained from each independent replicate experiment). Statistical significance was determined using an unpaired Welch's t test (B and E), a Mann-Whitney test (C), or by calculating Z scores (D and F; *, $p \leq 0.1$; **, $p \leq 0.05$; ***, $p \leq 0.005$). Also see *Figure 2—figure supplements 1*, *2*, *3* and *4*, and *Figure 2—source data 1*.

DOI: https://doi.org/10.7554/eLife.47246.005

The following source data and figure supplements are available for figure 2:

**Source data 1.** Spreadsheet with spindle dynamics assay values for haploid cells.
DOI: https://doi.org/10.7554/eLife.47246.010
**Figure supplement 1.** H3639P mutant exhibits synthetic genetic interactions with *KAR9*.
DOI: https://doi.org/10.7554/eLife.47246.006
**Figure supplement 2.** Additional plots of spindle dynamics data from haploid cells.
DOI: https://doi.org/10.7554/eLife.47246.007
**Figure supplement 3.** Mutations map to various structural elements within the motor domain.
DOI: https://doi.org/10.7554/eLife.47246.008
**Figure supplement 4.** Tail domain mutations cluster to two distinct regions.
DOI: https://doi.org/10.7554/eLife.47246.009

degrees of alterations in their motility parameters with respect to wild-type cells. At the most severe end of the spectrum, H3639P cells – which also had the most severe spindle positioning phenotype – exhibited only 10 dynein-mediated spindle displacement events from all cells observed (compare to a mean of 244 events for all other strains; see *Figure 2—figure supplement 2A and B* for scatter plots illustrating density of datasets). Consequently, this mutant had extremely low 'activity' parameters (*i.e.*, total spindle displacement, *Figure 2E*; and, number of dynein-mediated spindle movements per minute, *Figure 2F*). The mutant with the second most severe spindle positioning phenotype, R1852C, exhibited significant defects in all motility metrics, including velocity, displacement (per event), neck transit success, and the two activity parameters. The relative phenotypic severity of these two mutants is consistent with findings from another group in which disease-correlated dynein mutants were assessed using recombinant human dynein complexes (*Hoang et al., 2017*). This group identified the human equivalents of R1852C (R1962C) and H3639P (H3822P) as being the most severe loss-of-function mutants in their reconstituted motility assays (see Discussion).

Another noteworthy mutant was K1475Q, a substitution within the linker domain, the mechanical element that is responsible for the powerstroke (*Figure 2—figure supplement 3*). This mutation led to a reduction in spindle velocity, displacement, and also reduced the activity metrics of the motor. Given the position of this mutation, it may affect dynein activity by compromising linker remodeling during the priming or powerstroke movement of the linker. However, a recent study identified the equivalent human residue (R1567) as being at least partly required for the formation of an autoinhibited conformation of human dynein called the phi-particle (*Zhang et al., 2017*) (due to its resemblance to the Greek letter; *Amos, 1989*), thus raising the possibility that yeast dynein also adopts this conformation. Thus, the altered motility of this mutant may be a consequence of altered activity regulation (see below and Discussion).

All three mutations within the AAA3 module of the motor domain led to varying degrees of spindle motility alterations (D2439K, R2543K, and L2557M; *Figure 2—figure supplement 3*). Interestingly, the most striking defect we observed with L2557M was a reduction in the neck transit success rate (*Figure 2D*), which is the likely cause of the spindle positioning defect. This suggests that this mutation – which is buried within the large subdomain of AAA3, and makes hydrophobic contacts with a closely apposed helix – is likely affecting force generation by the motor.

All three microtubule-binding domain (MTBD) mutants exhibited fairly similar degrees and types of defects in effecting spindle movements (reductions in velocity, displacement, and neck transit success). Structural analysis revealed that all three mutations mapped to the surface of the MTBD that makes contacts with the microtubule (*Figure 2—figure supplement 3*). Given all three substitutions

lead to loss of a positive charge, it is likely that these mutations each lead to a reduction in the affinity of the motor for the negatively charged surface of the microtubule. The reduced activity metrics for R3152N and R3201N could thus be a reflection of a reduction in association kinetics of the mutants for the microtubule. This is supported by a study that found reduced microtubule binding for similar amino acid substitutions in a mouse dynein MTBD fragment (*Poirier et al., 2013*).

Although none of them led to significant spindle positioning defects, all of the tail domain mutations led to altered spindle motility parameters. Structural analysis revealed that most of these mutations (7 out of the 8) clustered to two distinct regions: (1) adjacent to, or within the N-terminal dimerization domain, or (2) at a surface that interfaces with the intermediate chain of a neighboring heavy chain in a two dynein:1 dynactin complex (*Figure 2—figure supplement 4*). This latter region was recently identified as being important to stabilize the binding of a second dynein complex to dynactin, and ensuring that all four heavy chains are properly aligned for efficient motility of the human dynein-dynactin complex (*Urnavicius et al., 2018*). Our data suggest that the ability to recruit two dynein complexes to dynactin is conserved in yeast, and that disrupting this complex can compromise force generation (*Figure 2D*) or activity (*Figure 2E and F*). Although the last tail domain mutation, W612C, is in close proximity to this latter region, it is sufficiently far from the contact point with the second dynein complex to suggest that it is likely affecting some other facet of dynein-dynactin function.

## Mutations exert dominant negative effects on spindle dynamics

Given the heterozygous nature of these mutations in affected patients, we wondered how the mutants would behave in the presence of a second, wild-type copy of dynein. It is currently unclear whether cells with two copies of *DYN1* (*e.g.*, wild-type and mutant) assemble dynein complexes comprised of two different copies (*e.g.*, wild-type/mutant heterodimers), or whether they are comprised of only one copy (*e.g.*, wild-type homodimers, or mutant homodimers). This latter phenomenon could be a consequence of co-translational dimerization (*Natan et al., 2017*). To distinguish between these two possibilities, we generated diploid yeast strains that contained one copy of a GFP-tagged dynein heavy chain (*DYN1-GFP*), and a second copy that was fused to an N-terminal affinity tag and a C-terminal HALO tag (*ZZ-DYN1-HALO*), the latter of which could be used to fluorescently label the motor (*Figure 3A*). Lysate from these cells was subjected to affinity chromatography, and subsequent to incubation with a red fluorescent HALO ligand (HALO-TMR), the bound protein was eluted and used in a single molecule imaging experiment. If heterodimers assemble within cells, we expected to observe dual-color labeled molecules (green and red); however, if only homodimers form, then we expected to observe only red molecules (see *Figure 3A*). Although we observed a small number of motile green molecules (0.9% of the total; likely due to contaminating Dyn1-GFP molecules in the protein preparation), and a single dual labeled molecule (0.3% of the total; *Figure 3B*, arrow), the vast majority of motile molecules (98.8%) were exclusively red, indicating that dynein very rarely, if ever, forms heterodimers (*Figure 3C*). This suggests a co-translational dimerization model for dynein complex assembly, similar to what has been observed for p53 (*Nicholls et al., 2002*).

Next, we wished to recapitulate the heterozygous nature of some mutations in our budding yeast system. To this end, we chose three mutants – E545V, R1852C, H3639P – and determined whether they could affect dynein activity in heterozygous diploid yeast strains. We mated haploid wild-type cells with haploid mutant cells to generate heterozygous diploid cells (*e.g.*, *DYN1/dyn1^R1852C*; *Figure 3D*). In addition to comparing dynein activity in these cells to that from homozygous wild-type cells (*i.e.*, *DYN1/DYN1*), we also compared them to hemizygous cells with only one copy of *DYN1* (*i.e.*, *DYN1/dyn1Δ*; *Figure 3E–I*, and *Figure 3—figure supplement 1*). Although homozygous wild-type cells exhibited similar dynein-dynactin activity to the hemizygotes, the velocity was somewhat reduced in the latter, suggesting a critical concentration of dynein is required for effecting maximal spindle velocity (*Figure 3F*, and *Figure 3—figure supplement 1*). Analysis of the heterozygous mutants revealed that all three exhibited partial loss-of-function phenotypes in almost all assays with respect to the hemizygous cells, indicating that they are all indeed dominant alleles. Taken together, these data indicate that homodimers of mutant dynein are sufficient to compromise the activity of wild-type homodimers within the cell.

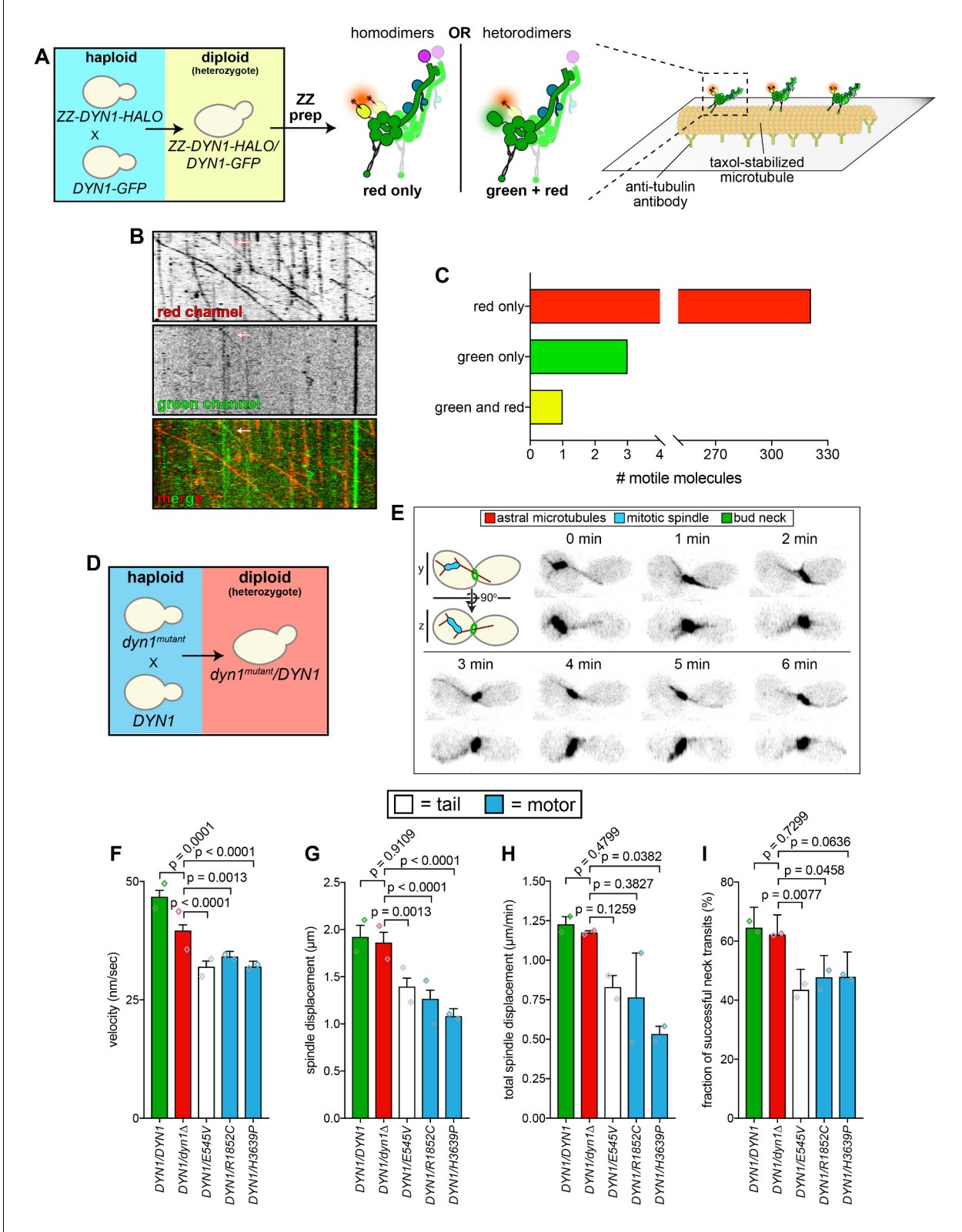

**Figure 3.** Quantitative assessment of spindle dynamics in heterozygous diploid cells reveals dominant nature of mutations. (**A**) Schematic depicting experimental approach to determine whether distinct proteins from two different dynein alleles homo- or heterodimerize. (**B and C**) Representative kymograph (**B**) depicting large proportion of red (HALO-tagged) dynein molecules walking along microtubules (only one of which colocalized with a GFP-tagged dynein; arrow), along with associated quantitation (**C**). (**D – E**) Schematic depicting experimental approach to assess dynein-dynactin

*Figure 3 continued on next page*

*Figure 3 continued*

activity in heterozygous diploid cells (D), and representative inverse fluorescence images of a diploid hydroxyurea (HU)-arrested *kar9Δ/kar9Δ* cell exhibiting typical dynein-mediated spindle movements. Maximum intensity (X-Y projection; top) and Y-Z projections (bottom) are shown for each time point. (F – I) Plots of indicated parameters for spindle dynamics in indicated diploid yeast strains. Each bar represents the weighted mean ±weighted standard error (or standard error of proportion for I; n ≥ 29 HU-arrested cells from two independent experiments were analyzed for each strain; diamonds represent mean values obtained from each independent replicate experiment). Statistical significance was determined using an unpaired Welch's t test (F and H), a Mann-Whitney test (G), or by calculating Z scores (I). Also see *Figure 3—figure supplement 1*, and *Figure 3—source data 1*.

DOI: https://doi.org/10.7554/eLife.47246.011

The following source data and figure supplement are available for figure 3:

**Source data 1.** Spreadsheet with spindle dynamics assay values for diploid cells.
DOI: https://doi.org/10.7554/eLife.47246.013
**Figure supplement 1.** Additional plots of spindle dynamics data from diploid cells.
DOI: https://doi.org/10.7554/eLife.47246.012

## Single molecule motility assays reveal insight into dynein-intrinsic dysfunction

Findings from our spindle tracking assay revealed the consequences of mutations on various parameters of dynein-dynactin-mediated spindle movements. These movements are mediated by cortically anchored dynein-dynactin complexes that are regulated at various levels by numerous effector molecules (*e.g.*, Pac1, Ndl1, Num1, She1; *Markus et al., 2012*; *Markus and Lee, 2011b*; *Li et al., 2005*; *Lammers and Markus, 2015*). For instance, cortical targeting of dynein is affected by various molecules, including Pac1, which tethers dynein to microtubule plus ends (*Lee et al., 2003*), and Num1, which anchors dynein-dynactin complexes to the cortex (*Heil-Chapdelaine et al., 2000*). Thus, mutations that alter the ability of dynein to interact with or be affected by these molecules will result in alterations in dynein-dynactin-mediated spindle movements. To determine whether mutations affect dynein-intrinsic activities, it is therefore important to specifically assess dynein activity without these complicating factors. To this end, we employed a single molecule motility assay, in which the movement of individual purified dynein motors is quantitatively assessed. Performing this assay using yeast dynein has one key advantage over human dynein: unlike human dynein, which requires dynactin and one of several adaptor molecules for processive single molecule motility (*e.g.*, BicD2, Hook3; *McKenney et al., 2014*; *Schlager et al., 2014*), yeast dynein is a processive motor without these factors (*Reck-Peterson et al., 2006*). This somewhat unique property of yeast dynein thus permits an unbiased assessment of dynein-intrinsic motility.

Single molecule motility analysis of each mutant revealed that thirteen of those that exhibited defects in the spindle tracking assay showed some degree of motility alteration in the in vitro assay (*Figure 4*; see *Figure 4—figure supplement 1* for scatter plots and some example kymographs). Interestingly, the precise defect in vivo was not always predictive of the alteration in dynein motility in vitro. Although the reasons for this are unclear, they are likely due in part to the differences in requirements for spindle transport versus those for unloaded single molecule motility (*i.e.*, in which there is no resistance to transport). Additionally, small defects in single motor motility may lead to more pronounced defects in the context of a motor ensemble, as may be the case for dynein at the cell cortex (*Markus et al., 2011*). Such mutants included W612C and R1852C, both of which led to a reduction in all spindle tracking metrics in cells, but had very little effect on the displacement (run length) of single molecules in vitro. Similarly, R2543K, which reduced spindle velocity by ~50% had only a minor effect on single molecule velocity values.

In a few cases, we observed little or no difference from wild-type in single molecule motility parameters in spite of differences in the spindle tracking assay (*i.e.*, E109I, K540C, E545V, I554M, D2439K, L2557M). In two of these cases (E545V and L2557M), one of these changes was a reduction in the neck transit success rate (see *Figure 2D*). As discussed above, this phenomenon is potentially a readout of force generation. Given the unloaded nature of single molecule motility, this assay would be unable to detect differences in force generation by dynein. Although the molecular determinants of dynein force production are not well understood, it is possible that these mutations specifically affect the ability of dynein to remain bound to microtubules under conditions of high load.

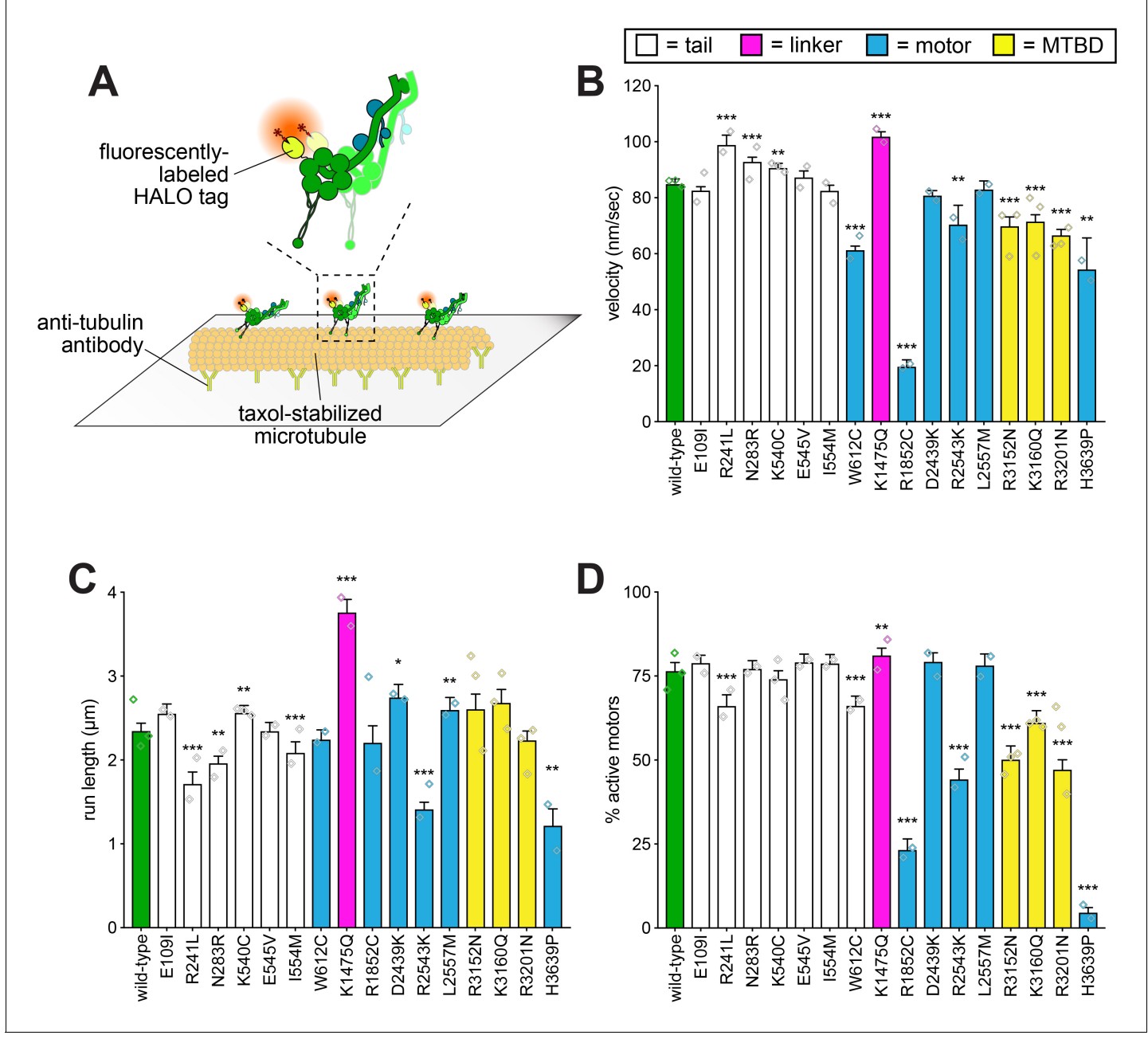

**Figure 4.** Single molecule analysis reveals insight into dynein-intrinsic dysfunction. (**A**) Cartoon representation of experimental approach. (**B – D**) Quantitation of indicated parameters of single molecule motility. Each bar represents the weighted mean ± weighted standard error (or standard error of proportion for D; n ≥ 284 single molecules from at least two experiments from independent protein preparations; diamonds represent mean values obtained from each independent protein preparation). Technical difficulties precluded us from generating the L213I mutant in the yeast strain used for protein purification. Statistical significance was determined using an unpaired Welch's t test (**B**), a Mann-Whitney test (**C**) or by calculating Z scores (**D**; *, p≤0.1; **, p≤0.05; ***, p≤0.005). Also see *Figure 4—figure supplements 1* and *2*, and *Figure 4—source data 1*.

DOI: https://doi.org/10.7554/eLife.47246.014

The following source data and figure supplements are available for figure 4:

**Source data 1.** Spreadsheet with all full-length and minimal GST-dimerized dynein single molecule motility values.
DOI: https://doi.org/10.7554/eLife.47246.017

**Figure supplement 1.** Additional plots of single molecule data and some representative kymographs.
DOI: https://doi.org/10.7554/eLife.47246.015

**Figure supplement 2.** K1475Q mutant does not form aggregates in single molecule assay.
DOI: https://doi.org/10.7554/eLife.47246.016

Although most mutants exhibited loss-of-function phenotypes in the in vitro assay, a few mutations led to gain-of-functions. For instance, R241L, N283R, and K1475Q all led to an increase in velocity, while K1475Q also caused an increase in run length and a small but significant increase in the fraction of active motors (see *Figure 4—figure supplement 1C* for example kymograph). We confirmed the increased run length for K1475Q was not a consequence of motor aggregates, which could presumably lead to an increase in apparent processivity (*Derr et al., 2012*) (*Figure 4—figure supplement 2*). As mentioned above, R241 and N283 are adjacent to the N-terminal dimerization domain, and K1475 is within the linker domain (see *Figure 2—figure supplements 3* and *4*). In the cases of R241L and N283R, the gain-of-functions observed in vitro are possibly causative of the in vivo deficiencies, indicating that a faster or more processive motor is not advantageous for the spindle positioning function (see below and Discussion regarding K1475Q). This is consistent with recent studies that showed gain-of-functions in dynein (or dynein regulators) can lead to defects in dynein-mediated neuronal maturation (*Huynh and Vale, 2017*), or spindle orientation (*Zhang et al., 2017*).

In summary, in all cases in which motility parameters were altered in vitro as a consequence of a particular mutation, we are able to conclude that the underlying molecular defect is most likely intrinsic to dynein itself, and is likely not a consequence of an altered interaction with either accessories or regulators.

## Localization phenotypes provide mechanistic basis for dysfunction in a subset of mutants

Our in vivo and in vitro functional data described above revealed the particular parameters of dynein motility that were altered by the mutations, and also whether the altered motility was in fact intrinsic to dynein (*i.e.*, if there was an in vitro phenotype), or was possibly a consequence of alterations in interactions with regulators such as dynactin. As discussed above, proper dynein function in yeast relies on the coordinated action of various molecules to localize dynein to microtubule plus ends, from where it is offloaded to its site of action: the cell cortex (*Figure 5A*) (*Markus and Lee, 2011b*). For instance, dynein plus end localization occurs in a dynactin-independent manner (*Moore et al., 2008*), but requires an interaction with Pac1 (the LIS1 homolog) (*Lee et al., 2003*), as well as the accessory chains Dyn3 (*Markus and Lee, 2011a*) (light-intermediate chain) and Pac11 (intermediate chain) (*Lee et al., 2005*). In contrast, dynactin is required for dynein to localize to cortical Num1 receptor sites (*Moore et al., 2008*). Thus, quantitative assessment of dynein localization can reveal potentially altered interactions with various regulators or accessories.

For this analysis, we focused on a select group of mutants: those with substitutions within the N-terminal tail domain (which is the site for interaction with the accessory chains, dynactin, and Num1; *Markus et al., 2009*), those with the most severe phenotypes (R1852C, R2543K, and H3639P), and two of the MTBD mutants. We acquired time-lapse images of haploid cells expressing fluorescently-labeled tubulin (mRuby2-Tub1; α-tubulin) and 3GFP-tagged copies of each dynein mutant. Plus end and cortical foci were identified from movies, and their targeting frequency (*i.e.*, percent cells with foci) and fluorescence intensities (a readout of molecule number per site) were quantified.

This analysis revealed significantly altered localization frequencies or intensities for most of the mutants, including L213I, N283R, E545V, I554M, W612C, K1475Q, R1852C, R2543K, and H3639P (*Figure 5B–D*). For instance, the K1475Q substitution (within the linker) led to a large increase in the frequency of observing cortical foci (2.2-fold; p=0.0011), but a concomitant reduction in the number of molecules per cortical focus (38% reduction in fluorescence intensity; p=0.0117), whereas W612C, R1852C and H3639P all exhibited a reduction in dynein levels at plus end and cortical sites (see Discussion). Finally, immunoblotting revealed that protein expression differences likely do not account for the observed localization or activity phenotypes (*Figure 5—figure supplement 2*).

The increased frequency of cortical foci for K1475Q suggests that this mutant may exhibit higher affinity for Pac1 and/or dynactin, the former of which is limiting for plus end association, and the latter of which is limiting for offloading to cortical Num1 sites (*Markus et al., 2011*). Given the importance of the human equivalent of K1475 (R1567) in stabilizing the autoinhibited 'phi' particle conformation – which exhibits lower affinity for dynactin than the open, uninhibited state (*Zhang et al., 2017*) – we wondered whether K1475Q interacts more readily with dynactin. To address this, we measured the relative ratio of dynein (using Dyn1-3GFP) to dynactin (Jnm1-3mCherry, homolog of p50/dynamitin) at microtubule plus ends, where dynactin recruitment is

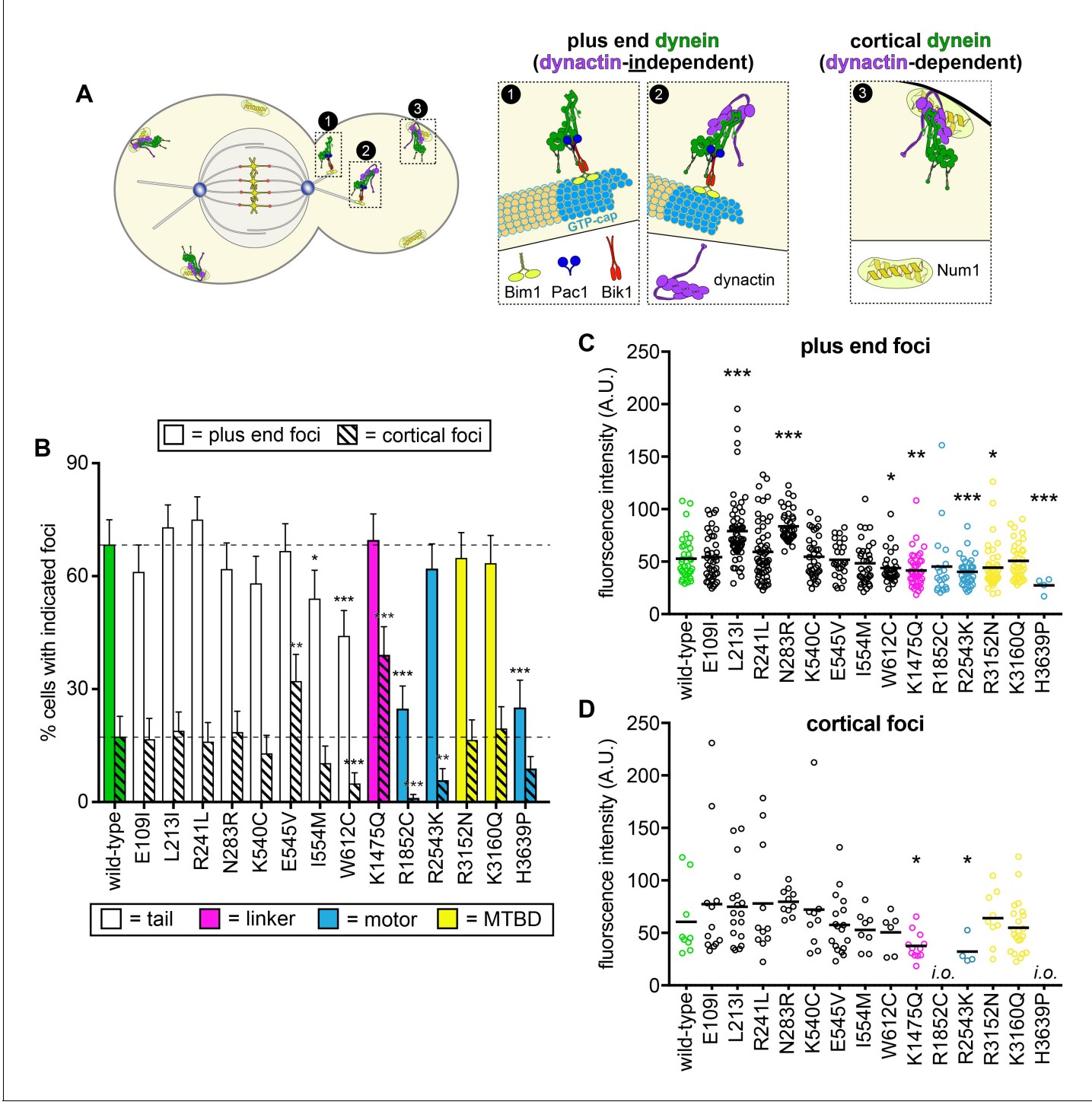

**Figure 5.** Quantitative assessment of dynein localization reveals potential basis for mutant dysfunction. (A) Cartoon representation depicting the two main sites of dynein localization, and the molecular basis for each. Dynein plus end localization (1) requires Bik1, Pac1, and possibly Bim1, but does not require dynactin. Rather, dynactin plus end localization (2) relies on dynein. Association of dynein with the cortex (3) requires dynactin and the cortical receptor, Num1. (B) The frequency of dynein localization to either microtubule plus ends or the cell cortex is plotted for indicated strains. To enrich for mitotic cells, overnight cultures were diluted into fresh media for 1.5 hr prior to imaging. To further reduce variability due to cell cycle-dependent changes (*Markus et al., 2009*), localization frequency was scored for mitotic cells only. Each data point represents the weighted mean ± weighted standard error (68 to 111 mitotic cells from at least two independent experiments were analyzed for each strain). (C and D) Fluorescence intensity values for either plus end (C) or cortical (D) dynein foci observed in mitotic cells described in B (4 to 59 plus end foci, and 4 to 22 cortical foci from two independent experiments were analyzed; *i.o.*, insufficient observations; only one cortical focus was observed for both R1852C and H3639P). Statistical

*Figure 5 continued on next page*

*Figure 5 continued*

significance was determined by calculating Z scores (**B**), or by applying an unpaired Welch's t test (**C and D**; *, p≤0.1; **, p≤0.05; ***, p≤0.005). Also see *Figure 5—figure supplements 1* and *2*, and *Figure 5—source data 1*.

DOI: https://doi.org/10.7554/eLife.47246.018

The following source data and figure supplements are available for figure 5:

**Source data 1.** Spreadsheet with localization frequency and intensity values for wild-type and mutant Dyn1-3GFP.

DOI: https://doi.org/10.7554/eLife.47246.021

**Figure supplement 1.** K1475Q mutant recruits more dynactin to microtubule plus ends than wild-type dynein.

DOI: https://doi.org/10.7554/eLife.47246.019

**Figure supplement 2.** Immunoblotting reveals no major differences in steady-state expression levels of dynein mutants.

DOI: https://doi.org/10.7554/eLife.47246.020

wholly dependent on dynein (*Moore et al., 2008*). Since assembled dynein-dynactin complexes are readily offloaded to cortical Num1 receptor sites (*Markus and Lee, 2011b*), we chose to perform this analysis in cells lacking Num1 (*num1Δ*) to exclude any contribution from the offloading process to the relative ratio of dynein to dynactin at plus ends (*Figure 5—figure supplement 1A*). Interestingly, this analysis indeed revealed a significantly increased ratio of dynactin to dynein at plus ends (*Figure 5—figure supplement 1B*; from a mean of 1.07 to 1.52; p<0.0001), indicating that K1475Q interacts more readily with dynactin than wild-type dynein, and that this mutation may in fact be disrupting a potential autoinhibited state of yeast dynein.

## Proline-dependent structural constraint is the likely cause for dysfunction in dynein H3639P

Given the wealth of structural information currently available for dynein, we sought to determine the structural basis for dysfunction in two of the mutants: R1852C and H3639P. Analysis of available dynein structures revealed that H3639 is situated within a conserved loop that connects two alpha helices, one of which is an extension of the buttress, a structural element that helps to communicate nucleotide-dependent structural rearrangements within the AAA ring to the MTBD (*Figure 6A*) (*Schmidt et al., 2015*). We first asked whether gain of proline or loss of histidine is the cause for the severe loss-of-function. To this end, we substituted H3639 with either a serine (to preserve the polar nature and hydrogen-bonding capabilities of histidine), valine (a hydrophobic residue), or asparagine (found in the equivalent position of human dynein-2), and assessed the activity of these mutants in the spindle positioning assay. None of these substitutions led to significant spindle positioning defects, indicating that gain of proline is the reason for dynein dysfunction in H3639P (*Figure 6B*). We next asked if proline substitutions are tolerated at other sites within this inter-helical loop. Using the spindle positioning assay, we found that proline was well tolerated at all sites within the inter-helical loop except for position 3641 (2 residues C-terminal to 3639; *Figure 6B*). Thus, introduction of a proline at two distinct sites within this loop are sufficient to severely compromise dynein function.

We hypothesized that prolines are not tolerated in these two regions (residues 3639 and 3641) of the inter-helical loop because of the structural constraints imposed by prolines due to bond angle restrictions. If this were true, then we reasoned that increasing the structural flexibility in the immediate vicinity of P3639 might rescue the proline-dependent defects. To test this, we introduced glycine substitutions at either the N-terminal residue (F3638), the C-terminal residue (W3640), or both. For reasons that are unclear, both double mutants (*i.e.,* F3638G H3639P, and H3639P W3640G) led to spindle positioning defects that were significantly more severe than H3639P and dynein knock-out cells (*dyn1Δ*; p<0.0001; *Figure 6B*). Strikingly, however, the triple mutant – in which P3639 is flanked by glycines (*i.e.,* F3638G H3639P W3640G) – exhibited spindle positioning defects that were significantly less severe than H3639P (p<0.0001).

To confirm these findings, we assessed the activity of the triple mutant using the spindle tracking assay. Although the activity of this mutant was much less than that of wild-type dynein-expressing cells, it was significantly greater than the H3639P single mutant (p≤0.0361; *Figure 6C–E*). In spite of the increase in activity, the quality of the spindle movements (velocity and displacement per event) were nearly identical between H3639P and the triple mutant (*Figure 6—figure supplement 1A–D*). Interestingly, the single molecule assay revealed that the triple mutant was no better than the single

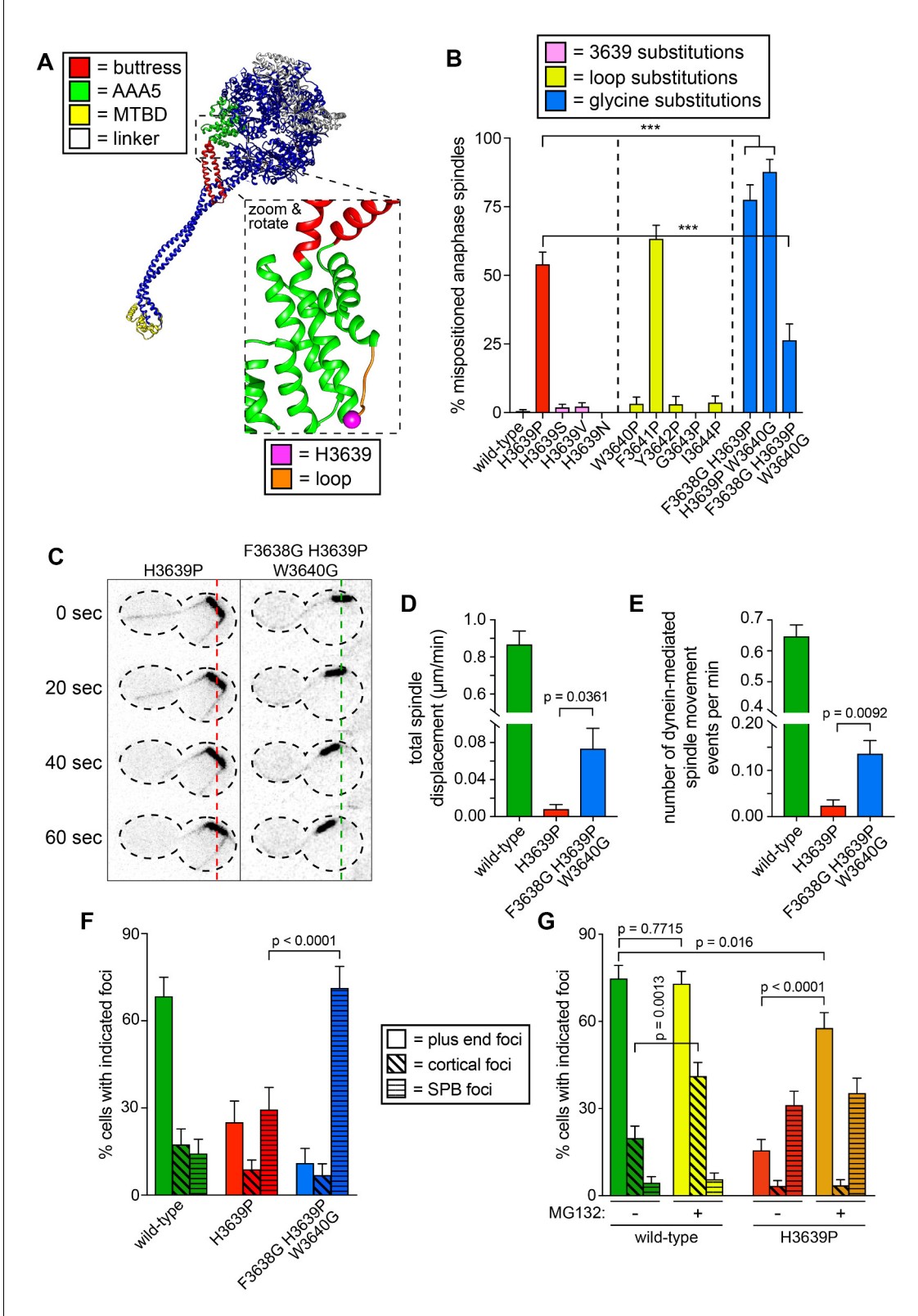

**Figure 6.** Detailed dissection of the molecular basis for dysfunction in H3639P. (**A**) Color-coded structural model of the dynein motor domain (from PDB 4RH7; *Schmidt et al., 2015*) with zoomed in region depicting H3639 residing within an inter-helical loop within AAA5. (**B**) Fraction of cells with mispositioned spindles are plotted for yeast strains with indicated dynein mutations. Each data point represents the fraction of mispositioned anaphase spindles along with standard error (weighted mean ± weighted standard error of proportion; n ≥ 67 anaphase spindles from at least two independent *Figure 6 continued on next page*

*Figure 6 continued*

experiments for each strain; ***, p≤0.0001). (**C**) Representative time-lapse inverse fluorescence images of two hydroxyurea (HU)-arrested *kar9Δ* cells with indicated dynein mutations. Note the lack of spindle translocation in H3639P, but the clear dynein-mediated movement in the F3638G H3639P W3640G mutant (dashed lines provide a point of reference). Maximum intensity projections are shown for each time point. (**D – E**) Plots of two activity parameters for spindle dynamics in indicated haploid strains (total displacement, **D**; and, number of events per minute, **E**). Each data point represents the weighted mean ± weighted standard error (**D**) or standard error of proportion (**E**; n ≥ 29 HU-arrested cells from at least two independent experiments were analyzed for each strain). (**F and G**) The frequency of dynein localization to either microtubule plus ends, the cell cortex, or spindle pole bodies (SPBs) is plotted for indicated strains and drug treatment (for mitotic cells only). In addition to the indicated alleles and drug treatment, the plot in panel **G** depicts cells that possess the *prd1-DBD-CYC8* allele, which represses transcription of pleiotropic drug resistance genes (*Stepanov et al., 2008*), thus promoting intracellular retention of MG132. These cells were treated with 75 µM MG132 for 1.5 hr prior to imaging (control cells were treated with an equal volume of DMSO). Each data point represents the weighted mean ± weighted standard error (73 to 107 mitotic cells from two independent experiments were analyzed for each strain). Statistical significance was determined by calculating Z scores (**B, E, F and G**), or by applying an unpaired t-test with Welch's correction (**D**). Also see *Figure 6—figure supplement 1*, *Figure 6—source datas 1* and *2*, and *Figure 4—source data 1*.

DOI: https://doi.org/10.7554/eLife.47246.022

The following source data and figure supplement are available for figure 6:

**Source data 1.** Spreadsheet with spindle dynamics assay values for inter-helical loop mutants.
DOI: https://doi.org/10.7554/eLife.47246.024
**Source data 2.** Spreadsheet with localization frequency values for wild-type and mutant Dyn1-3GFP in the absence and presence of MG132.
DOI: https://doi.org/10.7554/eLife.47246.025
**Figure supplement 1.** Additional insight into the molecular basis for dysfunction in H3639P.
DOI: https://doi.org/10.7554/eLife.47246.023

mutant in any metrics (*Figure 6—figure supplement 1E–I*). This indicates that the triple mutant does not rescue dynein motility, but some other metric of in vivo dynein (or dynein-dynactin) activity. We next performed live cell imaging to determine if the triple mutant alters any aspects of dynein localization. Although the flanking glycines did not rescue plus end or cortical localization, it did lead to a large increase in the fraction of cells exhibiting dynein foci at the spindle pole bodies (SPBs; *Figure 6F*; p<0.0001), suggesting that the triple mutant shifts the balance toward more semi-active – or properly folded – dynein within cells with respect to the single H3639P. The relevance of dynein localization to the SPB is unclear, but previous work from our lab demonstrate that this localization requires the MTBD (*Lammers and Markus, 2015*). In light of our other observations, the increased localization frequency of the triple mutant suggests that the flanking glycines rescue H3639P cellular defects by reducing proline-dependent inflexibility, which in turn may prevent global misfolding of Dyn1, thus increasing the relative concentration of active dynein within the cell. If H3639P causes some fraction of Dyn1 to be misfolded within the cell, then we reasoned that increasing the total amount of protein in cells would lead to an increase in the number of active Dyn1 molecules, and thus an increase in the apparent degree of H3639P localization. To test this, we assessed dynein localization in the absence or presence of MG132, a proteasome inhibitor, addition of which would lead to an increase in total protein content, including dynein. Addition of MG132 increased the frequency of wild-type cortical dynein foci by approximately 2-fold (p=0.0013), but had no significant effect on the frequency of either SPB or plus end targeting (*Figure 6G*). Consistent with our hypothesis, MG132 treatment increased the frequency of plus end targeting of H3639P by 3.7-fold (p<0.0001) to levels comparable to wild-type dynein; however, the frequency of cortical targeting of H3639P was unaffected by proteasome inhibition.

## An ectopic disulfide bond is the likely cause for R1852C dysfunction

Close inspection revealed a cysteine situated within very close proximity to R1852 (~3 Å; *Figure 7A*), which lies within the first AAA module (AAA1). Given the mutation results in a cysteine substitution at this site, we reasoned that an ectopic disulfide bond may be responsible for the phenotypic consequences. To determine whether this was the case, we mutated the closely apposed, highly conserved cysteine to a serine (C1822S; *Figure 7B*), which would eliminate the potential for disulfide bond formation at this site. We then used several of our assays to quantitate the degree to which C1822S might rescue defects due to R1852C.

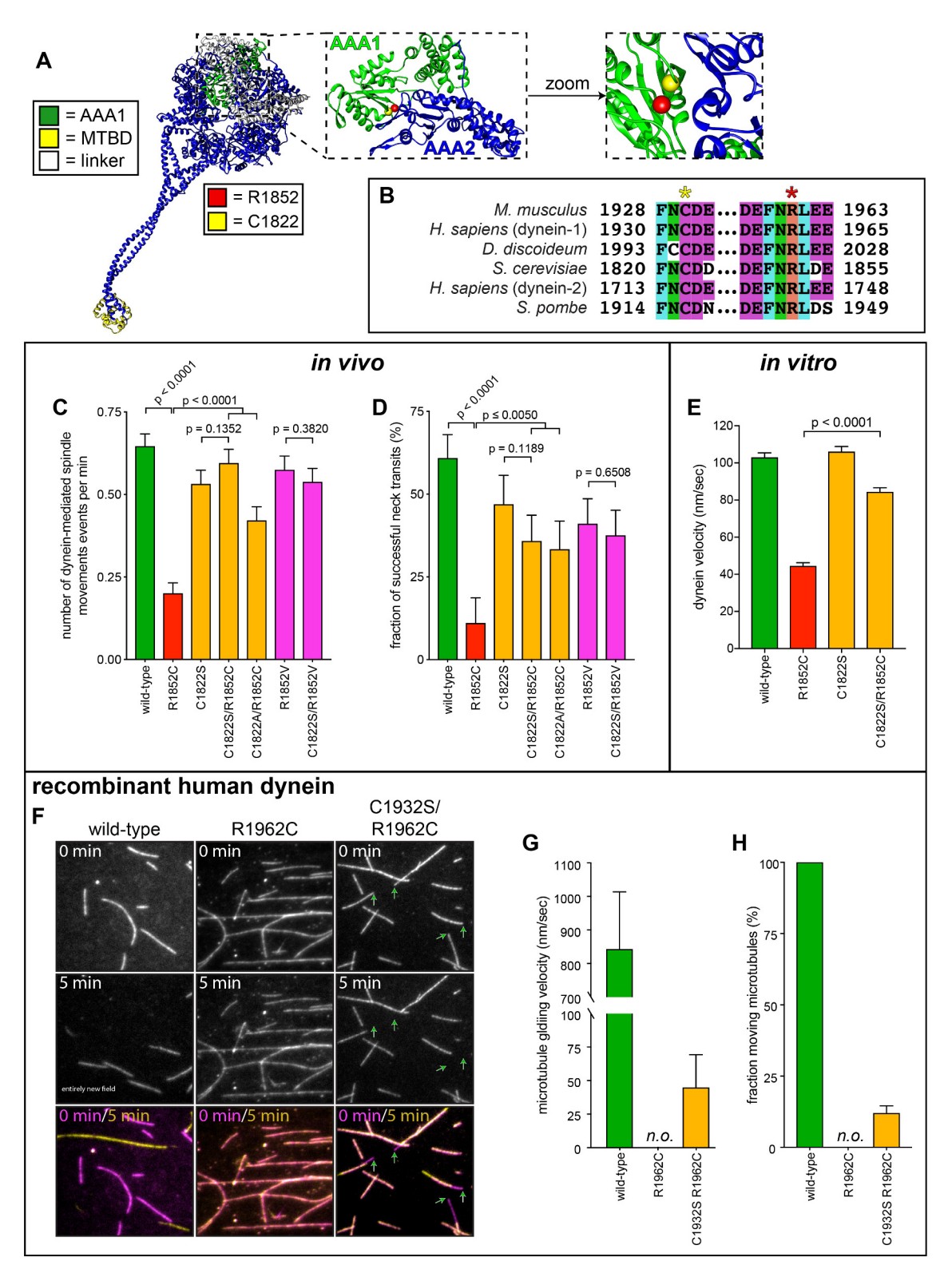

**Figure 7.** Detailed dissection of the molecular basis for dysfunction in R1852C. (**A**) Color-coded structural model of the dynein motor domain (from PDB 4RH7; *Schmidt et al., 2015*) with zoomed in region depicting R1852 residing within AAA1 near its interface with AAA2. (**B**) Sequence alignment illustrating the high degree of conservation for C1822 (yellow star) and R1852 (red star) among various dynein heavy chains. (**C and D**) Plots of two activity parameters for spindle dynamics in indicated haploid strains (total displacement, **C**; and, fraction of successful neck transits, **D**). Each data point

*Figure 7 continued on next page*

*Figure 7 continued*

represents the weighted mean ± weighted standard error (C) or standard error of proportion (D; n ≥ 28 HU-arrested cells from two independent experiments were analyzed for each strain). (E) Plot of velocity values for single molecules of indicated GST-dynein$_{331}$ variants (n ≥ 158 single molecules from at least two independent experiments were analyzed for each motor variant). (F) Representative fields of microtubules being translocated by surface-adsorbed recombinant human dynein complexes. (G and H) Plots depicting velocity (G) and fraction (H) of microtubules (at least 50 microtubules from two independent experiments were analyzed for each variant; '*n.o.*", none observed). Statistical significance was determined by calculating Z scores (B and E), or by applying an unpaired Welch's t test (D). Also see *Figure 7—figure supplements 1* and *2*, *Figure 7—source data 1*, and *Figure 4—source data 1*.

DOI: https://doi.org/10.7554/eLife.47246.026

The following source data and figure supplements are available for figure 7:

**Source data 1.** Spreadsheet with spindle dynamics assay values for R1852C related mutants.

DOI: https://doi.org/10.7554/eLife.47246.029

**Figure supplement 1.** Additional insight into the molecular basis for dysfunction in R1852C.

DOI: https://doi.org/10.7554/eLife.47246.027

**Figure supplement 2.** Additional plots of spindle dynamics and single molecule data.

DOI: https://doi.org/10.7554/eLife.47246.028

Although C1822S alone compromised all parameters of dynein-dynactin-mediated spindle movements, this substitution led to a partial restoration of most of the parameters in the R1852C mutant (*i.e.*, C1822S R1852C double mutant; *Figure 7C and D*, and *Figure 7—figure supplement 1A–C*). We observed a similar restoration of function in a C1822A R1852C double mutant. We ruled out the possibility that the partially hydrophobic nature of cysteine is the cause for the dysfunction in R1852C by assessing an R1852V mutant, which exhibited defects that were significantly less severe than R1852C. Moreover, combining C1822S with R1852V led to no degree of rescue with respect to the single R1852V mutant, which is in stark contrast to our observations with the double C1822S R1852C mutant (*Figure 7C and D*, and *Figure 7—figure supplement 1A–C*). We also found that C1822S rescued the localization of R1852C to plus ends and the cell cortex almost to wild-type levels (*Figure 7—figure supplement 1D*). As with the spindle tracking assay, although R1852V led to localization defects, the C1822S R1852V double mutant was almost identical to R1852V alone.

Before assessing the extent of rescue with the single molecule assay, we first tested whether using a minimal dynein fragment that is sufficient for processive motility (*Reck-Peterson et al., 2006*) would rescue any of the motility defects we observed with the R1852C mutant (GST-dynein$_{331}$). This fragment – a glutathione S-transferase (GST)-dimerized motor domain that exhibits motility parameters that are very similar to the full-length molecule (*Reck-Peterson et al., 2006*) – lacks the tail domain, and thus does not co-purify with or rely on any of the accessory chains. If protein misfolding is partly to blame for mutant dysfunction – as suggested from the single molecule (only 23.2% active motors; *Figure 4D*) and localization assays (severe reduction in localization frequency; *Figure 5B*) – then we reasoned that the simplicity and compact fold of GST-dynein$_{331}$ might rescue some of these defects. In striking contrast to the full-length mutant, the fraction of active GST-dynein$_{331}$ R1852C motors was almost identical to that of the wild-type GST-dynein$_{331}$ motor (note the same construct did not rescue the H3639P mutant; *Figure 7—figure supplement 1E*), suggesting that the holoenzyme complex is more susceptible to defects arising from this mutation than is the truncated motor domain. However, much like the full-length molecule, the velocity of the GST-dynein$_{331}$ mutant was severely reduced with respect to wild-type (*Figure 7E* and *Figure 7—figure supplement 1G*). Consistent with the spindle tracking data, C1822S was indeed sufficient to significantly rescue the single molecule velocity defect in the minimal dynein fragment (*Figure 7E* and *Figure 7—figure supplement 1G*). Taken together, these findings indicate that an ectopic disulfide bond is the likely cause for R1852C dysfunction.

## Compensatory mutation partially suppresses motility defects of human dynein mutant

Finally, we sought to determine whether the compensatory cysteine to serine substitution would also rescue motility defects in human dynein. To this end, we engineered the equivalent mutations (C1932S, R1962C, or C1932S R1962C) into an insect cell expression plasmid encoding an affinity-tagged human dynein complex (ZZ-SNAPf-DYNC1H1, IC2, LIC2, Tctex1, Robl1, LC8; a kind gift from

Andrew Carter). Subsequent to purification of recombinant complexes, we assessed their function using a microtubule gliding assay in which free microtubules are translocated by dynein complexes non-specifically adsorbed to the glass surface. This assay permits an assessment of dynein-intrinsic motility parameters without the need for additional factors that are required for single molecule motility of human dynein (*i.e.*, dynactin and a requisite adaptor [*McKenney et al., 2014*; *Schlager et al., 2014*]). Consistent with previous findings, the single R1962C mutation (equivalent to R1852C) was sufficient to completely disrupt microtubule gliding activity (*Hoang et al., 2017*) (*Figure 7F–H*); however, as with yeast dynein, substitution of the closely apposed cysteine to a serine (C1932S) was sufficient to restore some of the lost function. Although the compensatory mutation did not restore activity to wild-type levels (note the low velocity and degree of activity with respect to wild-type in *Figure 7G and H*), the same was true for the full-length yeast dynein complex, which was only partially rescued by the mutation (see *Figure 7C and D*, and *Figure 7—figure supplement 1A–C*). Although unclear, the different degrees of rescue by the compensatory mutation with yeast dynein (observed in vivo; *Figure 7C and D*, and *Figure 7—figure supplement 1A–D*) versus human dynein (observed in vitro; Fig. F-H) may be due to the presence of quality control mechanisms at play in live yeast cells which ensure that mostly properly folded motors are delivered to cortical receptor sites. No such mechanism is present to prevent adsorption of inactive recombinantly produced human dynein motors to the glass in the in vitro gliding assay. In summary, these results confirm our findings with yeast dynein, and moreover validates yeast as a powerful model system to understand the molecular basis for dynein dysfunction in patients.

## Severity of dynein dysfunction correlates with disease type

In an effort to identify potential correlations between degree of dynein dysfunction and disease, we summarized the findings from all of our various assessments into a heat map in which the variance (or lack thereof) from wild-type was assigned a color based on the statistical significance (*e.g.*, green, $p \geq 0.100$; red, $p \leq 0.005$; *Figure 8A*). Unsurprisingly, on average, mutations within the highly conserved motor domain exhibited more statistically significant differences from wild-type than those in the tail domain. Moreover, this broad view of dysfunction in the mutant library revealed that the more severe loss-of-function mutants appeared to correlate with MCD, while those with lesser defects mainly correlate with motor neuron diseases (SMA-LED, CMT or CMD).

To more quantitatively assess our data for potential correlations between dynein dysfunction and disease type, we developed a system in which the degrees of variance from wild-type for each mutant were assigned numerical scores that we then used to tabulate a single value that represents the degree of dynein dysfunction for each mutation (coefficient of dynein dysfunction, or CDD). For tabulation of the CDD value, we focused on the in vivo data only. The reasons for this were two-fold: (1) defects observed in the in vivo assays were largely reflective of the degree of dysfunction, whereas the in vitro data generally revealed the mechanisms of dysfunction (*e.g.*, whether the mutation affected dynein-intrinsic or extrinsic function); and, (2) all mutants with in vivo defects also exhibited in vitro defects, and so we chose to include only the in vivo data to avoid redundancies in the CDD tabulation. In addition to the spindle positioning assay, values from the following spindle dynamics metrics were used to compile the CDD: velocity, displacement per event, neck transit success rate, total spindle displacement per minute, and number of events per minute. The latter two were both included in the CDD tabulation since they revealed two different aspects of dynein activity: the extent to which the mutant dynein-dynactin complexes could move the spindle (total displacement per minute), and the ability to initiate a spindle movement event (number of events per minute). Since the spindle positioning assay reveals gross perturbations in dynein function, we increased the weight of this score with respect to each of the other metrics, such that the summed spindle dynamics metrics were weighed equally with that of the spindle positioning assay (see *Figure 8—figure supplement 1* for additional details on CDD tabulation). This analysis revealed a broad range of dynein dysfunction ranging from low (<10) to high CDDs (>30) for the mutant library. For comparison, the CDDs for wild-type and *dyn1Δ* cells were set to 0 and 100, respectively. Thus, considering the phenotypic severity of H3639P (CDD = 82) – which had a spindle positioning defect as severe as *dyn1Δ*, but exhibited some dynein activity in the spindle dynamics assays – we find that the CDD score accurately reflects the degree of dysfunction for each mutant.

Ranking the mutants according to their CDD scores (from low to high; *Figure 8B*) revealed an apparent correlation between dynein dysfunction and disease type. Specifically, we found that above

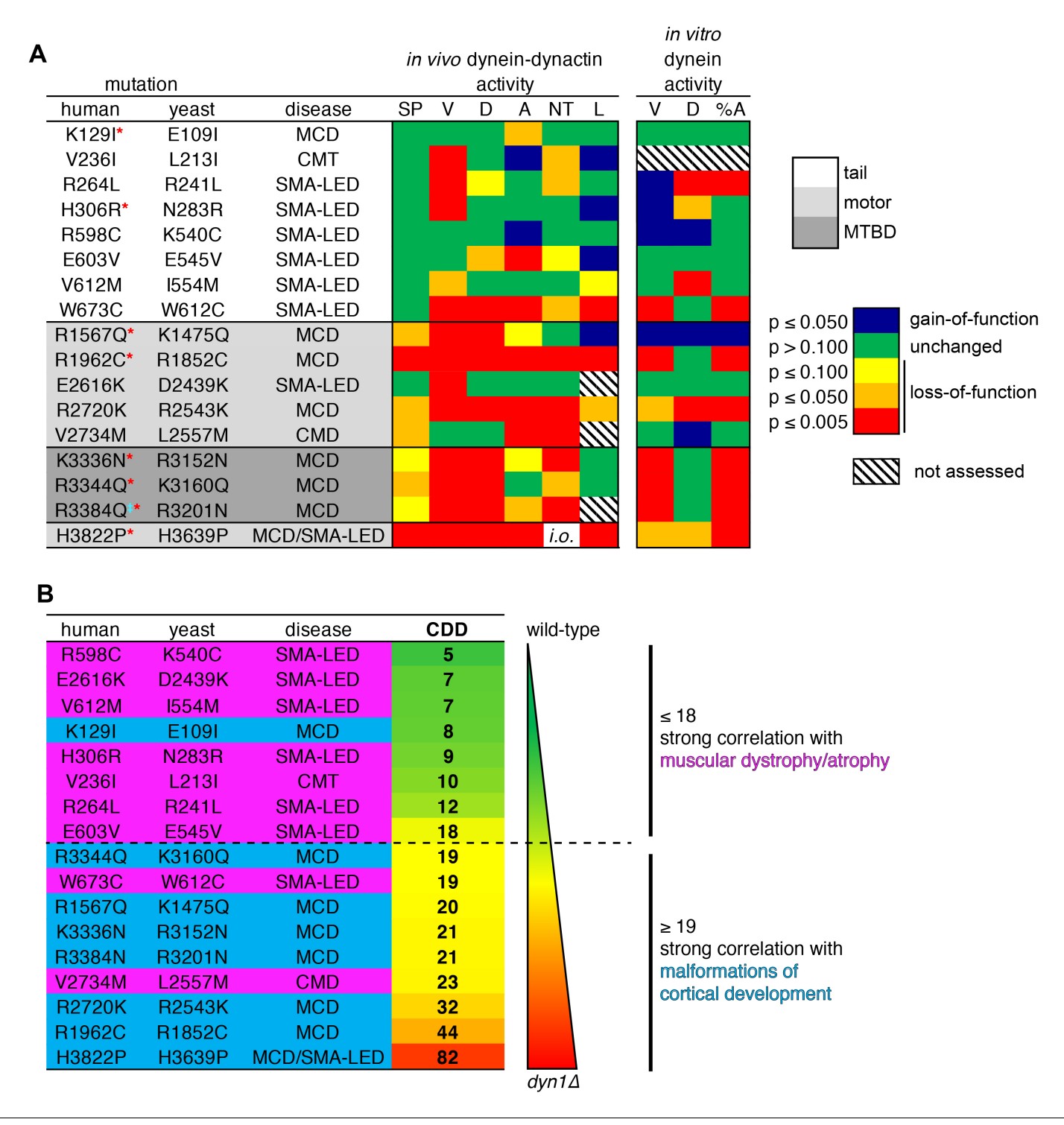

**Figure 8.** Summary of phenotypic analysis of disease-correlated mutants. (**A**) Heat map depicting degree of statistical significance for difference between each mutant and wild-type cells for the indicated assays (SP, spindle positioning; V, velocity; D, displacement per event; A, activity; NT, neck transit; L, localization; %A, percent active motors; MCD, malformations in cortical development; CMT, Charcot-Marie-Tooth disease; SMA-LED, spinal muscular atrophy with lower extremity dominance; CMD, congenital muscular dystrophy). Red asterisks depict mutants that were assessed in a previous study using recombinant human dynein (*Hoang et al., 2017*). ‡Note that we mistakenly substituted an asparagine for residue R3201 instead of a glutamine, the latter of which was identified as correlating with MCD (*Poirier et al., 2013*). R3201N was used throughout this study. Deviation from wild-type cells in either the 'total spindle displacement' (see *Figure 2E*), or the 'number of dynein-mediated spindle movements per minute' (see *Figure 8 continued on next page*

*Figure 8 continued*

*Figure 2F*) metric was used for the activity column ('A'). Mutants are listed from N- to C-terminus of Dyn1 (shading indicates in which domain of Dyn1 each mutation resides). Significance was calculated as indicated in previous figure legends. (B) Each mutant ranked by their coefficient of dynein dysfunction score (CDD; see text and *Figure 8—figure supplement 1*). As shown, above a CDD score of ~ 18, the correlation with malformations in cortical development increases, while that with motor neuron diseases decreases. Also see *Figure 8—figure supplement 1*.

DOI: https://doi.org/10.7554/eLife.47246.030

The following figure supplement is available for figure 8:

**Figure supplement 1.** Data used for tabulation of the coefficient of dynein dysfunction (CDD).

DOI: https://doi.org/10.7554/eLife.47246.031

a CDD value of ~ 18, the likelihood of the mutation correlating with MCD increased, while its correlation with SMA-LED, CMT or CMD decreased. This suggests that the two general types of diseases (motor neuron disease, and defects in brain development) are each caused by different degrees of dynein dysfunction, and that there exists a lower threshold of dynein activity that is required for neurological development (see Discussion).

## Discussion

Neurodegenerative or developmental diseases that arise as a consequence of mutations within the dynein gene – or dyneinopathies – are a broad range of devastating diseases that include muscular atrophy, muscular dystrophy, and malformations in cortical development (MCD). The ages of onset for these diseases range from birth to late adulthood, while the severity of the symptoms associated with them also cover a broad range (*Poirier et al., 2013*; *Laquerriere et al., 2017*; *Willemsen et al., 2012*; *Vissers et al., 2010*). Although the underlying reasons behind this symptomatic diversity are unclear, our findings suggest that the degree of dysfunction is at least a potential genetic determinant of the type of disease. Specifically, motor neuron diseases appear to be more susceptible to even small degrees of dysfunction (CDD from 5 to 18), while MCD tends to correlate with larger degrees of dynein dysfunction (CDD $\geq$ 19). The reasons for this are unclear, but we hypothesize that the differences are due to the somewhat distinct types of dynein-mediated transport required to maintain motor neuron health, versus that required to effect nuclear or neuronal migration in the developing neocortex. Motor neurons possess extremely long axons ($\leq$1 m), and thus require a high degree of processive transport for the myriad vesicular cargoes that are moved from the soma to the axon terminal and back. This transport takes place not only during development and in growing children, but also in fully matured adults. Thus, it stands to reason that even subtle loss-of-functions in dynein-mediated transport can, over time, compromise motor neuron health. This is apparent from the wide range in ages-of-onset (from birth to adulthood), and in the range of severity for dynein-based motor neuron diseases (from weakness in the lower limbs to gross motor difficulties). On the other hand, during early brain development, dynein plays key roles in interkinetic nuclear migration (INM) and neuronal migration during development of the neocortex (*Tsai et al., 2010*; *Hu et al., 2013*; *Tsai et al., 2007*). Nuclear envelope-anchored dyneins move the nucleus tens of microns from the basal to the apical surface of the neuroepithelium (*Hu et al., 2013*). In addition to being a shorter distance traveled compared to axonal transport in a motor neuron, the number of motors engaged with microtubules during a nuclear migration event is likely far greater. Immunofluorescence reveals dynein is present along most of the nuclear envelope surface (*Hu et al., 2013*). Moreover, live cell imaging revealed that the microtubule network is fairly extensive in the proximity of the nucleus (*Tsai et al., 2010*; *Tsai et al., 2007*). A similar process may be at play during post-INM neuronal migration, when dynein assists in centrosome advancement of the postmitotic neuron. Evidence suggests that, as in budding yeast, astral microtubules make contacts with teams of cortically anchored dynein motors to move the centrosome (*Tsai et al., 2007*). Thus, for both INM and neuronal migration, many dynein molecules are likely engaging with microtubules to effect neurogenesis. This is in contrast to a vesicle that is being transported along the axon of a motor neuron, which likely possesses far fewer motors (~3–7 per vesicle [*Hendricks et al., 2010*; *Rai et al., 2013*]), of which potentially only a subset are engaged due to the geometric constraints associated with a three-dimensional vesicle engaging with a single filament. Thus, the minimal functional requirements for individual dynein motors during INM or neuronal migration are potentially

lower due to the large number of motors engaged during a migration event. In such a scenario, teams of motors comprised of a mixture of wild-type and mutant variants (due to the heterozygous nature of these diseases) can work together to effect INM, but are less able to effectively transport single vesicular cargoes along the axon.

Our data indicate that K1475Q – a mutation in the linker domain – increases dynein run length in vitro, and alters its localization pattern in vivo. The position of this mutation (at an intermolecular interface that helps mediate formation of an autoinhibited Phi particle conformation of human dynein; *Zhang et al., 2017*) suggests that the associated phenotypes may be a consequence of altered activity regulation. Given this mutant exhibited dynein-intrinsic enhancements in in vitro activity, the reduction in cellular dynein-dynactin activity is likely a consequence of the altered localization pattern. Specifically, the localization phenotype raises the possibility that a reduction in the number of dynein-dynactin complexes per cortical site (as apparent by the reduced fluorescence intensity of cortical foci; *Figure 5D*) – which would result in fewer motor complexes being engaged for a spindle movement event – is the basis for cellular dysfunction. Given the autoinhibited Phi particle exhibits reduced affinity for dynactin (*Zhang et al., 2017*), the altered localization pattern of the mutant may be due to disruption of a similar autoinhibitory conformation in yeast, consequent increased dynactin binding – which is supported by our ratiometric fluorescence imaging (*Figure 5— figure supplement 1*) – and thus an increase in the frequency of cortical off-loading events (*Markus and Lee, 2011b*). Previous studies have suggested that dynein's interaction with dynactin is a limiting step in the delivery of dynein-dynactin complexes to cortical Num1 sites (*Markus et al., 2011*). We are currently focused on assessing whether yeast dynein indeed adopts such an autoinhibited state, and what role this conformation plays in the regulation of dynein activity.

In addition to revealing the potential molecular basis for disease onset or progression in affected patients, our findings also identified the potential structural basis for dynein dysfunction in two of the mutants. In the case of H3639P, our data support a model wherein the proline substitution compromises structural plasticity within an inter-helical loop in AAA5 that ultimately leads to a loss of activity in a large fraction of the motors, potentially as a consequence of protein misfolding. This conclusion is based on the reduced localization phenotype (*Figure 5B and C*) that is rescued by flanking glycines (*Figure 6F*) and proteasome inhibition (*Figure 6G*), and the large proportion of non-motile microtubule-bound motors we observed in the single molecule assay (*Figure 4D*). Since nearly all these motors exhibit persistent microtubule binding but no motility (96%; see kymograph in *Figure 4—figure supplement 1*), we propose that a structural defect within the AAA ring is compromising the ability of nucleotide binding or hydrolysis to communicate with the microtubule-binding domain. It is interesting to note that the small fraction of motors that do exhibit processive motility in vitro (4.5%) move at velocities that are roughly similar to wild-type motors (80 versus 58 nm/sec for wild-type and H3639P, respectively). We observed a similar phenomenon during dynein-mediated spindle translocation in vivo (42 versus 22 nm/sec). These findings are consistent with the notion that a small fraction of the motors are capable of overcoming folding defects to adopt a native, motility-competent conformation.

Subsequent to initiating this study, another group published findings describing the in vitro motility properties of a subset of the mutants analyzed here (*Hoang et al., 2017*). In this study, the authors utilized a recombinant human dynein in complex with dynactin and the adaptor BicD2 to assess single molecule motility parameters. With only two exceptions (E109I and N283R), the findings from this highly informative study largely corroborate our own data in that the dynein mutants assessed compromised the processivity of dynein-dynactin complexes (see *Figure 8A*, left, 'D'). For instance, the two mutants with the most severe phenotypic consequences were the same as those observed here (R1962C and H3822P). Although one of the exceptions – E109I – did not exhibit reduced processivity in our assays, the authors' observation that this mutation leads to a reduction in the number of processive motors (*Hoang et al., 2017*) is similar to our observation of reduced cellular dynein-dynactin activity for this mutant (*Figure 8A*, left, 'A'). Interestingly, this same study also noted a similar correlation between degree of dynein dysfunction and disease type (*i.e.*, the most severe phenotypes correlated with MCD, while the least severe phenotypes correlated with SMA-LED), providing further validation for budding yeast as a model system for dynein studies.

In summary, we have established yeast as a medium-throughput model system that can be used to assess the molecular basis for dysfunction of disease-correlated dynein mutants. Our rapid and economical toolbox can be easily applied to understand the underlying basis for dysfunction in newly

identified dynein mutants found in patients suffering from neurological diseases. We have also demonstrated the feasibility of rapidly testing hypotheses generated from our battery of assays using the wealth of available structural information that is available for dynein and its regulators.

## Materials and methods

### Media and strain construction

Strains are derived from either W303 or YEF473A (*Bi and Pringle, 1996*) and are listed in *Supplementary file 1*. We transformed yeast strains using the lithium acetate method (*Knop et al., 1999*). Strains carrying mutations were constructed by PCR product-mediated transformation (*Longtine et al., 1998*) or by mating followed by tetrad dissection. Proper tagging and mutagenesis was confirmed by PCR, and in most cases sequencing (all point mutations were confirmed via sequencing). Fluorescent tubulin-expressing yeast strains were generated using plasmids and strategies described previously (*Markus et al., 2015*; *Song and Lee, 2001*). Yeast synthetic defined (SD) media was obtained from Sunrise Science Products (San Diego, CA).

### Plasmid generation

For expression and purification of human dynein complex mutants (or wild-type), mutations were engineered into the human dynein heavy chain (DHC)-containing plasmid, pbiG1a:6His-ZZ-SNAPf-DHC1. We used Gibson assembly to engineer point mutations – C1932S, R1962C, or both – into this plasmid. The PmeI-digested gene expression cassette from this plasmid was co-assembled with the PmeI-digested poly-gene cassette from pbiG1b:IC2/LIC2/Tctex1/Robl1/LC8 (encoding all dynein accessory chains) into PmeI-digested pbiG2ab using biGBac cloning strategies as previously described (*Weissmann et al., 2016*) (all wild-type plasmids were kind gifts from Andrew Carter). The final, sequence-verified plasmids (wild-type and mutant variants of pbiG2ab:6His-ZZ-SNAPf-DHC1/IC2/LIC2/Tctex1/Robl1/LC8) were used to generate recombinant baculoviral genomes by Tn7 transposition into DH10Bac cells (Life Technologies). White, PCR-confirmed colonies were inoculated into LB media supplemented with 7 µg/ml gentamycin, 10 µg/ml tetracycline and 50 µg/ml kanamycin and grown overnight at 37°C. Bacmid preparation was performed as described previously (*Zhang et al., 2017*), stored at 4°C, and used within 2 weeks for subsequent virus production (see below).

### Protein purification

Purification of yeast dynein (ZZ-TEV-Dyn1-HALO, under the native *DYN1* promoter; or, ZZ-TEV-6His-GFP-3HA-GST-dynein$_{331}$-HALO, under the control of the galactose-inducible promoter, *GAL1p*) was performed as previously described (*Ecklund et al., 2017*; *Huang et al., 2012*). Briefly, yeast cultures were grown in YPA supplemented with either 2% glucose (for full-length dynein) or 2% galactose (for GST-dynein$_{331}$), harvested, washed with cold water, and then resuspended in a small volume of water. The resuspended cell pellet was drop frozen into liquid nitrogen and then lysed in a coffee grinder (Hamilton Beach). After lysis, 0.25 vol of 4X dynein lysis buffer (1X buffer: 30 mM HEPES, pH 7.2, 50 mM potassium acetate, 2 mM magnesium acetate, 0.2 mM EGTA) supplemented with 1 mM DTT, 0.1 mM Mg-ATP, 0.5 mM Pefabloc SC (concentrations for 1X buffer) was added, and the lysate was clarified at 22,000 x g for 20 min. The supernatant was then bound to IgG sepharose six fast flow resin (GE) for 1–1.5 hr at 4°C, which was subsequently washed three times in 5 ml lysis buffer, and twice in TEV buffer (50 mM Tris, pH 8.0, 150 mM potassium acetate, 2 mM magnesium acetate, 1 mM EGTA, 0.005% Triton X-100, 10% glycerol, 1 mM DTT, 0.1 mM Mg-ATP, 0.5 mM Pefabloc SC). To fluorescently label the motors for single molecule analyses, the bead-bound protein was incubated with either 6.7 µM HaloTag-AlexaFluor660 (Promega) for 10 min at room temperature. The resin was then washed four more times in TEV digest buffer, then incubated in TEV buffer supplemented with TEV protease for 1–1.5 hr at 16°C. Following TEV digest, the beads were pelleted, and the resulting supernatant was aliquoted, flash frozen in liquid nitrogen, and stored at −80°C.

The human dynein complex was expressed and purified from insect cells (ExpiSf9 cells; Life Technologies) as previously described with minor modifications (*Zhang et al., 2017*; *Schlager et al., 2014*). Briefly, 4 ml of ExpiSf9 cells at 2.5 × 10$^6$ cells/ml, which were maintained in ExpiSf CD Medium (Life Technologies), were transfected with 1 µg of bacmid DNA (see above) using

ExpiFectamine (Life Technologies) according to the manufacturer's instructions. 5 days following transfection, the cells were pelleted, and 1 ml of the resulting supernatant (P1) was used to infect 300 ml of ExpiSf9 cells ($5 \times 10^6$ cells/ml). 72 hr later, the cells were harvested (2000 x g, 20 min), washed with phosphate buffered saline (pH 7.2), pelleted again (1810 x g, 20 min), and resuspended in an equal volume of human dynein lysis buffer (50 mM HEPES, pH 7.4, 100 mM NaCl, 10% glycerol, 1 mM DTT, 0.1 mM Mg-ATP, 1 mM PMSF). The resulting cell suspension was drop frozen in liquid nitrogen and stored at −80℃. For protein purification, 30 ml of additional human dynein lysis buffer supplemented with cOmplete protease inhibitor cocktail (Roche) was added to the frozen cell pellet, which was then rapidly thawed in a 37℃ water bath prior to incubation on ice. Cells were lysed in a dounce-type tissue grinder (Wheaton) using $\geq$ 150 strokes (lysis was monitored by microscopy). Subsequent to clarification at 40,000 x g, 45 min, the supernatant was applied to 2 ml of IgG sepharose fast flow resin (GE) pre-equilibrated in human dynein lysis buffer, and incubated at 4℃ for 2–4 hr. Beads were then washed with 50 ml of human dynein lysis buffer, and 50 ml of human dynein TEV buffer (50 mM Tris pH 7.4, 150 mM potassium acetate, 2 mM magnesium acetate, 1 mM EGTA, 10% glycerol, 1 mM DTT, 0.1 mM Mg-ATP). The bead-bound protein was incubated with 3 µM SNAP-Surface Alexa Fluor 647 (NEB) for 40–60 min at 4℃ (to fluorescently label the protein), washed five times in human dynein TEV buffer, then incubated with TEV protease overnight at 4℃. The next morning, the recovered supernatant was applied to a Superose six gel filtration column (GE) equilibrated in GF150 buffer (25 mM HEPES pH 7.4, 150 mM KCl, 1 mM MgCl$_2$, 5 mM DTT, 0.1 mM Mg-ATP) using an AKTA Pure. Peak fractions (determined by UV 260 nm absorbance and SDS-PAGE) were pooled, concentrated, aliquoted, flash frozen, then stored at −80℃.

## Cell lysis and immunoblotting

Yeast cultures were grown to similar mid-log phase densities (OD600 ~ 2) in 4 ml SD media, and harvested. Cell pellets were resuspended in 0.2 ml of 0.1 M NaOH and incubated for 10 min at room temperature as described (*Kushnirov, 2000*). Following centrifugation, the resulting cell pellet was resuspended in sample buffer. Equal amounts of total cell lysate (as determined from cell density prior to lysis) were loaded into each lane, transferred to PVDF and probed with a monoclonal anti-GFP antibody (at 1:250; Abm) followed by an HRP-conjugated goat anti-mouse antibody (at 1:10,000; Jackson ImmunoResearch Laboratories). Electroblotting to PVDF was performed in 25 mM Tris, 192 mM glycine supplemented with 0.05% SDS and 20% methanol. Chemiluminescence signal was acquired with a Chemidoc MP (BioRad). Immunoblots were exposed (durations ranged from 2 to 5 min) without saturating the camera's pixels.

## Single and ensemble molecule motility assays

The yeast dynein single-molecule motility assay was performed as previously described with minor modifications (*Ecklund et al., 2017*). Briefly, flow chambers constructed using slides and plasma cleaned and silanized coverslips attached with double-sided adhesive tape were coated with anti-tubulin antibody (8 µg/ml, YL1/2; Accurate Chemical and Scientific Corporation) then blocked with 1% Pluronic F-127 (Fisher Scientific). Taxol-stabilized microtubules assembled from unlabeled and fluorescently-labeled porcine tubulin (10:1 ratio; Cytoskeleton) were introduced into the chamber. Following a 5–10 min incubation, the chamber was washed with dynein lysis buffer (see above) supplemented with 20 µM taxol, and then purified dynein motors were introduced in the chamber. After a 1 min incubation, motility buffer (30 mM HEPES pH 7.2, 50 mM potassium acetate, 2 mM magnesium acetate, 1 mM EGTA, 1 mM DTT, 1 mM Mg-ATP) supplemented with 0.05% Pluronic F-127, 20 µM taxol, and an oxygen-scavenging system (1.5% glucose, 1 U/ml glucose oxidase, 125 U/ml catalase) was added. TIRFM images were collected using a 1.49 NA 100X TIRF objective on a Nikon Ti-E inverted microscope equipped with a Ti-S-E motorized stage, piezo Z-control (Physik Instrumente), and an iXon X3 DU897 cooled EM-CCD camera (Andor). 488 nm, 561 nm, and 640 nm lasers (Coherent) were used along with a multi-pass quad filter cube set (C-TIRF for 405/488/561/638 nm; Chroma) and emission filters mounted in a filter wheel (525/50 nm, 600/50 nm and 700/75 nm; Chroma). We acquired images at 2 s intervals for 8 min. Velocity and run length values were determined from kymographs generated using the MultipleKymograph plugin for ImageJ (http://www.embl.de/eamnet/html/body_kymograph.html).

Human dynein-mediated microtubule gliding assays were performed as previously described (*Zhang et al., 2017*) with minor modifications. Briefly, flow chambers were prepared by affixing an ethanol-flamed coverslip to a glass slide using double-stick tape. The chamber was then incubated on an ice block, washed with 1% Pluronic F-127, following by addition of purified dynein (five chamber volumes of 60 nM dynein complex). Unbound motors were washed out with GF150 buffer. Subsequently, motility buffer (30 mM HEPES pH 7.0, 50 mM KCl, 5 mM MgSO4, 1 mM EGTA, 1 mM DTT, 2.5 mM Mg-ATP, 40 µM taxol) supplemented with 1.5% glucose, the oxygen scavenging system (see above), and 150 nM fluorescent microtubules was added to the chamber. Images were acquired every 1 s (for wild-type) or 5 s (for mutants), and velocity values were determined from kymographs generated as above.

## Live cell imaging experiments

For the single time point spindle position assay, the percentage of cells with a misoriented anaphase spindle was determined after growth overnight (12–16 hr) at a low temperature (16℃), as previously described (*Li et al., 2005*; *Markus et al., 2009*; *Sheeman et al., 2003*). A single z-stack of wide-field fluorescence images was acquired for mRuby2-Tub1. For the spindle dynamics assay, cells were arrested with hydroxyurea (HU) for 2.5 hr, and then mounted on agarose pads containing HU for fluorescence microscopy. Full Z-stacks (23 planes at 0.2 µm spacing) of GFP-labeled microtubules (GFP-Tub1) were acquired every 10 s for 10 min on a stage pre-warmed to 30℃. To image dynein localization in live cells, cells were grown to mid-log phase in SD media supplemented with 2% glucose, and mounted on agarose pads. Images were collected on a Nikon Ti-E microscope equipped with a 1.49 NA 100X TIRF objective, a Ti-S-E motorized stage, piezo Z-control (Physik Instrumente), an iXon DU888 cooled EM-CCD camera (Andor), a stage-top incubation system (Okolab), and a spinning disc confocal scanner unit (CSUX1; Yokogawa) with an emission filter wheel (ET525/50M for GFP, and ET632/60M for mRuby2; Chroma). Lasers (488 nm and 561 nm) housed in a LU-NV laser unit equipped with AOTF control (Nikon) were used to excite GFP and mRuby2, respectively. The microscope was controlled with NIS Elements software (Nikon).

## Statistical analyses

Statistical tests were performed as described in the figure legends. Unpaired Welch's t tests (for gaussian distributed velocity data) and Mann-Whitney test (for exponentially distributed dispalcement data) were performed using Graphpad Prism. Z scores, which are a quantitative measure of difference between two proportions, were calculated using the following formula:

$$Z = \frac{(\hat{p}_1 - \hat{p}_2)}{\hat{p}(1-\hat{p})(\frac{1}{n_1} + \frac{1}{n_2})}$$

where:

$$\hat{p} = \frac{y_1 + y_2}{n_1 + n_2}$$

Z scores were converted to two-tailed P values using an online calculator.

## Coefficient of dynein dysfunction (CDD) score calculation

To calculate the CDD scores, we used the following approach which permitted a quantitative measure of difference between mean values obtained for wild-type versus those obtained for each mutant. Graphpad Prism was used to calculate q values (*i.e.*, the difference between the two means divided by the standard error of that difference), whereas Z scores were calculated as described above (all values are shown in *Figure 8—figure supplement 1*). We then converted the q values and Z scores for each mutant (for each assay) into a 'normalized relative variance' score (nrv), which reflects the relative difference between two mean values (*e.g.*, between wild-type and mutant 1; as reflected in the Z scores and q values, or 'v'), where nrv = |v|/$v_{max}$ for each range of scores (for each column shown in *Figure 8—figure supplement 1A*). To convert the nrv values into a final CDD score for each mutant, we used the formula shown in *Figure 8—figure supplement 1C*. Briefly, the nrv values for each assay for a given mutant was added, with the spindle positioning nrv (nrv$_{SP}$) weighed five times that of the others, as described within the Results. In the two cases where a value wasn't

determined (due to insufficient observations, such as in the case for neck transit success for the H3639P mutant), the denominator was reduced from 6 to 5. In the two instances where the Z score for spindle positioning was negative (due to a lower number of mispositioned spindles being observed in K540C and D2439K cells than in wild-type cells; see *Figure 1B*), we adjusted the values to 0 so as to avoid them skewing the $nrv_{SP}$ values.

## Acknowledgements

We are grateful to Andrew Carter for sharing unpublished reagents (the human dynein complex expression plasmids) and valuable discussions. We also want to thank Richard Vallee, James Bamburg, Ashok Prasad, Olve Peersen, and members of the Markus and DeLuca laboratories for valuable discussions. This work was funded by the Muscular Dystrophy Association (376387 to SMM) and the NIH/NIGMS (GM118492 to SMM). We also thank Dr. Jeffrey Moore (and NIH R01-GM112893) for providing support for CPF who developed Matlab code used for tracking spindles in live cells, and for sharing the *pdr1-DBD-CYC8* yeast strain.

## Additional information

### Funding

| Funder | Grant reference number | Author |
| --- | --- | --- |
| Muscular Dystrophy Association | 376387 | Matthew G Marzo<br>Jacqueline M Griswold<br>Kristina M Ruff<br>Rachel E Buchmeier<br>Steven M Markus |
| National Institute of General Medical Sciences | GM 118492 | Matthew G Marzo<br>Steven M Markus |
| National Institute of General Medical Sciences | GM 112893 | Colby P Fees |

The funders had no role in study design, data collection and interpretation, or the decision to submit the work for publication.

### Author contributions

Matthew G Marzo, Formal analysis, Validation, Investigation, Visualization, Methodology, Writing—review and editing; Jacqueline M Griswold, Formal analysis, Investigation, Visualization; Kristina M Ruff, Rachel E Buchmeier, Reagent building; Colby P Fees, Software, Developed and refined code used for 3-dimensional tracking of mitotic spindles in live cells; Steven M Markus, Conceptualization, Resources, Data curation, Supervision, Funding acquisition, Validation, Investigation, Writing—original draft, Project administration, Writing—review and editing

### Author ORCIDs

Matthew G Marzo ![iD] https://orcid.org/0000-0002-2571-6377
Steven M Markus ![iD] https://orcid.org/0000-0002-3098-0236

### Decision letter and Author response

Decision letter https://doi.org/10.7554/eLife.47246.037
Author response https://doi.org/10.7554/eLife.47246.038

## Additional files

### Supplementary files

• Source code 1. Matlab Code for tracking spindles in three-dimensions. To be used with *Source code 2*.
DOI: https://doi.org/10.7554/eLife.47246.032

• Source code 2. Supplementary Matlab Code for spindle tracking. To be used with *Source code 1* (required for defining threshold of fluorescence images).
DOI: https://doi.org/10.7554/eLife.47246.033
• Supplementary file 1. Yeast strains used throughout this study.
DOI: https://doi.org/10.7554/eLife.47246.034
• Transparent reporting form
DOI: https://doi.org/10.7554/eLife.47246.035

## Data availability

All of the data generated or analysed during this study are included in the manuscript and supporting files. Source data files have been provided for all figures.

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
