## [Decision Letter]

Thank you for submitting your article "Molecular basis for dyneinopathies reveals insight into dynein regulation and dysfunction" for consideration by *eLife*. Your article has been reviewed by four peer reviewers, and the evaluation has been overseen by a Reviewing Editor and Anna Akhmanova as the Senior Editor. The following individuals involved in review of your submission have agreed to reveal their identity: Richard J McKenney (Reviewer #2); Giampietro Schiavo (Reviewer #3).

The reviewers have discussed the reviews with one another and I have drafted this decision to help you prepare a revised submission.

Summary:

Your manuscript represents a tour-de-force analysis of cytoplasmic dynein mutations that have been previously discovered in human patients. You use a budding yeast system to perform in vivo and in vitro analysis of a panel of dynein mutants that cause various human maladies. This study nicely builds on previous work from Bullock lab (Hoang et al.) by combining in vitro measurements, which largely agree with the previously published data, with a range of in vivo assays to assess particular cellular defects associated with each mutation. We felt think this paper will be broadly interesting to readers and will serve as a resource and example of how to approach the characterization of dynein mutations discovered in the future.

The reviewers were supportive of publishing the work in *eLife* after the following essential revisions have been addressed.

Essential revisions:

1) Were the single molecule data in Figure 4 derived from one protein preparation per variant? If so, data needs to be provided that supports the reproducibility of results between different preparations. For example, the authors could present data comparing the consistency of values for three wild-type dynein preps, or for a subset of mutants. The authors should state how many preps were tried with each mutant and each assay.

2) The authors need to compare the expression levels of the different mutations in yeast. Can they rule out that some of the mutations affect protein levels?

3) The authors should verify the amount of yeast dynactin (NIP100) associated to the different mutants via direct immunoprecipitation. These results would greatly help them to reach firm conclusions regarding the points raised in the third paragraph of the Discussion.

Other points:

1) In Figure 7G the degree of rescue for the double mutant is actually very low (note the split y-axis). The authors should make this clearer in the text and discuss the possible reasons.

2) Poirier et al. (2013) described R3384Q. Is R3384N, which is modeled by Marzo et al. in the current study, also a human mutation?

3) It would be helpful for the reader to see directly which mutations were tested in the 2017 study. This could, for example, be added in the graphic in Figure 8A.

4) Including the amino acid positions for the human mutations in parentheses in Figure 1A would make it easier to cross reference studies.

5) Figure 3B. The identity of each channel should be labeled in the figure for clarity.

6) 'Stabilized' is spelled incorrectly in Figure 4A.

7) Figure 2 legend: state what i.o. means.

8) "Moreover, combining C1822S with R1852V led to no degree of rescue". Do the authors mean compared to each of the single mutants? This could be clarified.

9) "Compensatory mutation rescues motility of human dynein mutant". It would be more accurate to say that the mutation suppresses or partially suppresses the mutant phenotype.

10) For the parameters in Figure 2 what are the typical values for the full dynein mutant? This information is not essential but if it is available it would help the reader judge the scale of the effects caused by the disease mutations.

11) The authors should be careful with color usage in Figure 1A (and all figures). Some of the dynein domains are the same color as the annotated mutations.

12) Is the neck-transit arrow pointing in the correct direction in Figure 2A, top panel? The movement in the images appears opposite to the arrow.

13) The observation of potential co-translational dimerization is intriguing. The authors could cite and discuss the recent finding of DynAPs in the assembly of ciliary dyneins: https://elifesciences.org/articles/38497.

14) The color-coding of mutations in the bar charts is helpful and should be displayed in all figures. For instance it is missing in Figures 3 and 4.

15) The authors fail to discuss prior literature on the characterized dynein mutations that cause similar phenotypes in other organisms such as mice (i.e. LOA, CRA, SWA mutations).

16) The increased run-length for K1475Q is intriguing. The authors should measure single molecule intensities and compare to WT motors to be sure this isn't due to aggregation.

17) I don't understand the conclusion that the full-length R1852C mutant motors are aggregated/misfolded but the minimal motor construct is not. More explanation is needed here. Why would the mutation in the motor domain more strongly affect the full-length motor versus the truncated motor?

18) Did the authors try adding in different concentrations or types of reducing agents to their in vitro assays to determine if an ectopic disulfide bond really is to blame for the phenotypes?

19) It’s hard to follow the logic of why the authors jumped to the conclusion that a protein degradation response was to blame for the defects in H3639P mutant cells?

20) It would be more transparent to plot most of the data points either directly on to the bar charts or replace the bar charts with box and whisker plots.

21) The Abstract states there is no correlation between mutations and disease state, but then the authors find one. Should the first statement be "There is no known correlation [...]"?

22) Figure 1: the authors should add the meaning of CC and DD in the legend of panel A and use the green color to indicate the bar for K1475Q (mutation in the linker domain) in panel B.

23) The authors should try and remove some redundant wording (e.g. subsection “Single molecule motility assays reveal insight into dynein-intrinsic dysfunction “: the content of the parentheses in the first paragraph "e.g. Pac.1..." is redundant in light of the following two paragraphs) and modify some sentences (e.g. the kymograph in Figure 4—figure supplement 1 readily displays an increase in speed of K1475Q motors; however, it is difficult, if not impossible, to evince from this figure an increase in run length or in the fraction of active motors).

24) Likewise, the authors should consider being a little bit more cautious in some of their conclusions (e.g. it is almost impossible to rule out a contribution of adaptors and accessory factors to the phenotype observed in mammalian cells) or speculations (geometric constrains limiting the amount of motor complexes engaged in transport of a vesicle – an organelle is almost never perfectly round, which the exception of synaptic vesicles which are assembled after transport at synapses).

25) Would it be possible to remove the term i.o. (insufficient observations) from some of the Figures (e.g. Figure 5) by increasing the number of observations or at least state the number of observed events in the figure legend?

26) The readability of panels B and F of Figure 7 should be improved.

27) The CDD calculation method should be included in the Material and methods section of the main text.

28) The authors should consider including in the reference section some early references demonstrating the key role of cytoplasmic dynein in the nervous system of Metazoa (e.g. Saxton's lab, Drosophila; Fisher's lab, mouse), and some key reviews discussing the links between deficits in axonal transport and nervous system disorders, and the complexity of the clinical phenotypes caused by mutations in the DYNC1H1 gene.

29) Please justify the decision of using one-way ANOVA OR unpaired t-test OR Z-scores for panels of the same figure (e.g. Figure 2). To the readers, these panels are indistinguishable in nature (they are derived from the same experimental dataset). This comment applies to the majority of main and supplemental figures.

[Editors' note: further revisions were requested prior to acceptance, as described below.]

Thank you for resubmitting your work entitled "Molecular basis for dyneinopathies reveals insight into dynein regulation and dysfunction" for further consideration at *eLife*. Your revised article has been favorably evaluated by Anna Akhmanova as the Senior Editor, a Reviewing Editor, and two reviewers.

The manuscript has been improved but there are some remaining issues that need to be addressed before acceptance, as outlined below:

1) Address the following comments about toning down the claims of novelty.

Some of the claims about the novelty of the finding that there is a correlation between disease type and strength of the mutations' effects on dynein activity are too strong, and need to be toned down.

For example, the authors write in the Abstract "In addition to revealing molecular insight into dynein regulation, our data reveal an unexpected correlation between the degree of dynein dysfunction and disease type".

The last part of the sentence is misleading as Hoang et al., 2017, reached a similar conclusion:

"Interestingly, of the 16 human or mouse mutations that support production of the recombinant dynein complex, the 6 with the strongest effects on dynein motility are associated with MCD in humans (Figure 6). All three SMALED mutations that could be assayed in the context of the purified motor complex fell into the class with the weakest effects (Figure 6). These mutations are among the group that only compromised run lengths of processive dynein-dynactin-BICD2N complexes. These findings raise the possibility that MCD and SMALED are caused by different degrees of inhibition of dynein motility, rather than inactivation of distinct dynein-related processes".

Whilst the confirmation and extension of this correlation by Marzo et al. valuable, it would seem uncharitable not to mention that it had been reported before (and is therefore not unexpected to find this in the current study).

The authors could, for example, write in the Abstract that "our data strengthen the evidence that different disease types arise from different degrees of dynein function", and must include an acknowledgement in the main text (e.g. in the Discussion) that Hoang et al. proposed this previously based on their observations.

There is a similar statement in the last paragraph of the Introduction that also needs toning down.

2) It was not clear to us from the legends to the new Figure 5—figure supplement 2 what the two images for each blot represent. Are these different blots of the same extracts, or from two different extracts per experiment? This information should be included in the legend.

---

## [Author Response]

Essential revisions:1) Were the single molecule data in Figure 4 derived from one protein preparation per variant? If so, data needs to be provided that supports the reproducibility of results between different preparations. For example, the authors could present data comparing the consistency of values for three wild-type dynein preps, or for a subset of mutants. The authors should state how many preps were tried with each mutant and each assay.

The single molecule data displayed in Figure 4 are derived from at least two independent protein preparations, that were each used to acquire replicate single molecule data. We apologize for not including this information in our previous draft, which has now been added to the Figure 4 legend. To address the reviewers’ concern about reproducibility between replicate data sets, we have now added data points to the plots in Figure 4 that represent the mean values measured from each replicate (see diamonds in Figure 4B-D). In addition to adding these data points to Figure 4, we have also added them to plots in Figures 2 (depicting the spindle dynamics assay in haploid cells) and 3 (depicting the spindle dynamics assay in diploid cells). We thank the reviewer for the suggestion.

2) The authors need to compare the expression levels of the different mutations in yeast. Can they rule out that some of the mutations affect protein levels?

We have spent a considerable amount of time and effort over the last 2 months trying to develop optimal immunoblotting conditions for consistent and reproducible data that we could use to quantitate relative expression levels of each dynein mutant. Likely as a consequence of dynein’s large mass and low expression levels, this has proven somewhat difficult. In particular, we have encountered difficulties obtaining consistent electroblotting efficiency to PVDF membrane. That being said, we have now included results (see Figure 5—figure supplement 2) from three independent immunoblots. These data indicate that the steady-state cytoplasmic levels of each dynein mutant is roughly similar. As part of our troubleshooting, we performed Ponceau S staining of several of these membranes, which revealed differences in transfer efficiency across each membrane (and from membrane to membrane). Unfortunately, it was difficult to identify an emerging pattern, as sometimes the transfer worked best in the middle of the membrane, and at other times closer to the edges. Thus, we have not included a loading control (which would need to be run on a separate gel and transferred separately), since such a control (e.g., tubulin) would not reveal the extent of transfer inefficiencies across each membrane. We attempted to scan and quantitate the Ponceau S-stained membrane, but the low degree of staining (as a consequence of the low percent acrylamide gel – 5% – and the somewhat harsh transfer conditions required to electroblot dynein to the PVDF) made it difficult to sufficiently detect the protein bands.

We have updated the manuscript to include these findings. We thank the reviewers for the suggestion.

3) The authors should verify the amount of yeast dynactin (NIP100) associated to the different mutants via direct immunoprecipitation. These results would greatly help them to reach firm conclusions regarding the points raised in the third paragraph of the Discussion.

As suggested by the reviewer, we sought to provide additional support for conclusions discussed in our manuscript. In particular: “Given the auto-inhibited Phi particle exhibits reduced affinity for dynactin, the altered localization pattern of the mutant may be due to disruption of a similar autoinhibitory conformation in yeast, consequent increased dynactin binding, and thus an increase in the frequency of cortical off-loading events.”Thus, if the K1475Q mutant indeed perturbs an autoinhibited conformation of dynein, then we would expect this mutant to exhibit an enhanced interaction with dynactin.

To test this, we chose to use a fluorescence microscopy-based method. The reasons for this were two-fold: (1) obtaining purified dynactin has thus far proven elusive (we have spent more than two years trying to obtain pure dynactin following methods outlined in Kardon et al., 2009 PNAS, and we are unable to reproduce these results); and, (2) we have spent much time and effort trying to develop a coimmunoprecipitation protocol, and the IP has either failed, or the protein of interest (e.g., Nip100-HA) bound non-specifically to the beads.

Thus, to determine whether the K1475Q mutant binds more readily to dynactin, we sought to measure the relative degree of dynein-mediated recruitment of dynactin to microtubule plus ends. Previous studies have shown that dynactin is directly recruited to microtubule plus ends by dynein (Moore et al., 2008 Traffic) but dynein does not require dynactin to associate with plus ends. Moreover, our own quantitative fluorescence microscopy-based measurements revealed that dynactin is limiting at plus ends with respect to dynein (~3 dyneins: 1 dynactin; Markus et al., 2011). Thus, if K1475Q disrupts an autoinhibited conformation of dynein, then we predict that this mutant would recruit more dynactin to plus ends than wild-type dynein. We performed ratiometric measurements of dynein (Dyn1-3GFP) and dynactin (using Jnm1-3mCherry) in cells deleted for Num1 in order to prevent the offloading mechanism from complicating interpretation of our measurements. These data, which are presented in Figure 5—figure supplement 1, indeed reveal an increased ratio of dynactin to dynein^K1475Q^ with respect to wild-type dynein (from 1.07 to 1.52; p < 0.0001). We thank the reviewer for the suggestion.

Other points:1) In Figure 7G the degree of rescue for the double mutant is actually very low (note the split y-axis). The authors should make this clearer in the text and discuss the possible reasons.We have added a note in the text clarifying the low degree of rescue, and also discussed a possible reason for the difference between human and yeast rescue. We thank the reviewer for the suggestion.2) Poirier et al. (2013) described R3384Q. Is R3384N, which is modeled by Marzo et al. in the current study, also a human mutation?

Thanks to the reviewer’s comment, we now realize we mistakenly generated R3201N instead of R3201Q (R3384 equivalent). Given the amount of time and work it would take to generate the R3201Q yeast mutants (for single molecule motility assays, and live cell imaging), and the amount of work that would be required to acquire and analyze data with R3201Q (and respective wild-type replicates), we propose leaving the data in with the caveat explicitly stated that the mutation that we analyzed is very similar to, but not identical to the one identified in the Poirer study.

We think the data are still valuable for the following reasons: (1) glutamine and asparagine differ from each other by only one methyl group, and are similarly distinct in their biochemical properties from arginine (both result in the loss of a positive charge); (2) all three microtubule-binding domain (MTBD) mutants (R3152N, K3160Q and R3201N) lead to similar phenotypes in the in vitro and in vivo activity assessments (see Figure 8A), suggesting that any mutation that leads to altered charge density at the microtubule-MTBD interface leads to a similar phenotype; (3) although the mutation is not necessarily modeled after a mutation found in patients, we think it provides valuable information pertaining to the importance of the native residues at this surface, and the consequences associated with mutations in this region.

We have added comments addressing this mistake to the legends for Figures 1 and 8 (from Figure 1 legend: “Note that we mistakenly substituted an asparagine for residue R3201 instead of a glutamine, the latter of which was identified as correlating with MCD. R3201N was used throughout this study.”) We thank the reviewer for pointing out this mistake.

3) It would be helpful for the reader to see directly which mutations were tested in the 2017 study. This could, for example, be added in the graphic in Figure 8A.As suggested by the reviewer, we have added red asterisks to the table in Figure 8A denoting which mutants were tested in the 2017 Hoang et al. (PNAS 2017) study. We thank the reviewer for the suggestion.4) Including the amino acid positions for the human mutations in parentheses in Figure 1A would make it easier to cross reference studies.As suggested, we have noted the human amino acid residues corresponding to the equivalent yeast residues in Figure 1A. We thank the reviewer for the suggestion. Please note that we have also indicated in this figure the mistaken amino acid substitution discussed above in point 2 (denoted with a ‡).5) Figure 3B. The identity of each channel should be labeled in the figure for clarity.This has been corrected. We thank the reviewer for pointing out this missing label.6) 'Stabilized' is spelled incorrectly in Figure 4A.This has now been corrected. We thank the reviewer for pointing out this mistake.7) Figure 2 legend: state what i.o. means.This has been added. We thank the reviewer for pointing out this oversight.8) "Moreover, combining C1822S with R1852V led to no degree of rescue". Do the authors mean compared to each of the single mutants? This could be clarified.We indeed meant with respect to the R1852V single mutant. This has been clarified in the text. We thank the reviewer for pointing out this point of confusion.9) "Compensatory mutation rescues motility of human dynein mutant". It would be more accurate to say that the mutation suppresses or partially suppresses the mutant phenotype.We agree with the reviewer’s suggestion, and have changed the header accordingly.10) For the parameters in Figure 2 what are the typical values for the full dynein mutant? This information is not essential but if it is available it would help the reader judge the scale of the effects caused by the disease mutations.As mentioned above, we have now included data points on the plots in Figure 2 (as well as those in Figures 3 and 4) that represent the mean values measured for each independent replicate. Moreover, we have also included scatter plots for the velocity and displacement datasets to display the entirety of the data (see Figure 2—figure supplement 2B and C).11) The authors should be careful with color usage in Figure 1A (and all figures). Some of the dynein domains are the same color as the annotated mutations.We apologize for the confusing color usage in Figure 1A. We have changed the coloring in this figure to avoid overlap of mutations with domain labeling. With the exception of Figures 6 and 7 – which focus on detailed characterizations of H3639P and R1852C, respectively – we employed the tail (white), motor (blue), linker (magenta), and MTBD (yellow) standard defined in Figure 1A for the remaining figures. We hope the reviewer finds this to be sufficiently improved.12) Is the neck-transit arrow pointing in the correct direction in Figure 2A, top panel? The movement in the images appears opposite to the arrow.To clarify, the arrow was not meant to depict the direction of the neck transit, but the time frames over which the neck transit was occurring. We changed the arrows to lines, and clarified this point in the legend to Figure 2. We thank the reviewer for pointing out this confusing labeling.13) The observation of potential co-translational dimerization is intriguing. The authors could cite and discuss the recent finding of DynAPs in the assembly of ciliary dyneins: https://elifesciences.org/articles/38497.

It is indeed intriguing to speculate that assembly of cytoplasmic dynein complexes occurs in a manner analogous to that described in the Huizar et al., 2018 eLife paper. The authors of this study provide compelling evidence demonstrating that axonemal dynein complexes assemble in liquid-like organelles within the cytoplasm with the aid of axonemal dynein assembly factors. However, our findings only suggest that dimerization of the heavy chain occurs in a co-translational manner, and does not suggest anything with regard to dynein complex assembly occurring in a discrete organelle. In fact, the assembly of axonemal dynein complexes in the DynAPs was not observed to take place coincident with translation, or even in the presence of RNAs from which the complex components may be translating, which contrasts with what we observe in budding yeast.

An example that is more analogous to what we have observed for DYN1 would be what was noted to occur with p53 (Nicholls et al., JBC 2002). In this study, the authors observed co-translational assembly of p53 dimers, but not higher-order (i.e., tetramers) oligomers. We have included this example in the latest iteration of our manuscript.14) The color-coding of mutations in the bar charts is helpful and should be displayed in all figures. For instance it is missing in Figures 3 and 4.We have now ensured that the color usage defined in Figure 1A (tail, white; motor, blue; linker, magenta; MTBD, yellow) is used throughout all figures, with the exception of Figures 6 and 7, which focus on detailed characterizations of H3639P and R1852C, respectively. We thank the reviewer for suggesting as much.15) The authors fail to discuss prior literature on the characterized dynein mutations that cause similar phenotypes in other organisms such as mice (i.e. LOA, CRA, SWA mutations).We have now included a brief discussion on the findings describing neuronal deficits in mouse models of dynein dysfunction (in the Introduction). We thank the reviewer for the suggestion.16) The increased run-length for K1475Q is intriguing. The authors should measure single molecule intensities and compare to WT motors to be sure this isn't due to aggregation.As suggested by the reviewer, we have now measured the fluorescence intensity values for single molecules of motile wild-type and K1475Q dynein. As shown in the new Figure 4—figure supplement 2, we did not observe any indication of motor aggregates in the single molecule assay. We thank the reviewer for the good suggestion.17) I don't understand the conclusion that the full-length R1852C mutant motors are aggregated/misfolded but the minimal motor construct is not. More explanation is needed here. Why would the mutation in the motor domain more strongly affect the full-length motor versus the truncated motor?This is an excellent question. We speculated that the truncated motor domain would be less susceptible to defects arising from the R1852C mutation (which, our data indicate are structural in nature) due the compact and simple nature of this fragment (i.e., does not associate with, or rely on accessory chains). That being said, the fact that the truncated R1852C mutant motor is more active than the full-length molecule is still a bit surprising to us. Without additional information, it is difficult to speculate further. However, we added a statement saying our data suggest “that the holoenzyme complex is more susceptible to defects arising from this mutation than is the truncated motor domain.” We are still actively exploring this question, although it is unclear we will ever know the precise reason the truncation rescues activity of the R1852C mutant.18) Did the authors try adding in different concentrations or types of reducing agents to their in vitro assays to determine if an ectopic disulfide bond really is to blame for the phenotypes?

In an attempt to reverse the disulfide bond within the R1852C mutant, we have indeed tested varying concentrations of DTT and N-acetyl cysteine in our in vivo and in vitro assays. Specifically, we tried adding 5-10 mM DTT (for the duration of the hydroxyurea arrest: 2.5 hours) to R1852C mutant cells prior to imaging for the spindle dynamics assay, but observed no effect. We also tried adding 1-10 mM DTT, and 1-10 mM N-acetyl cysteine (in separate experiments) for as long as two hours prior to imaging R1852C mutant cells to try to rescue defects in the localization assay, but observed no effect. Finally, we tested up to an additional 5 mM DTT to the single molecule assay with R1852C to try to restore motility, but observed no effect.

One simple explanation to account for the inability of the reducing agents to rescue the defects associated with R1852C is that the region of interest is not solvent accessible. Analysis of high resolution structures of the yeast dynein motor domain (pdb 4AKG) and human dynein-2 motor domain (pdb 4RH7) reveals this may be the case, as C1822S (or its human equivalent) is not apparent in either surface-rendered structure.19) It’s hard to follow the logic of why the authors jumped to the conclusion that a protein degradation response was to blame for the defects in H3639P mutant cells?We apologize for the lack of clarity in this section. We originally came to this conclusion based on the following pieces of evidence: (1) the frequency of cells exhibiting Dyn1^H3639P^ foci is low with respect to wild-type cells (suggesting lower amounts of protein); (2) this frequency can be increased by flanking Pro3639 with glycines (suggesting increased levels with respect to H3639P); (3) this triple mutant rescues only one aspect of dynein function: the relative number of dynein-mediated spindle movement events (not the quality of the events; note it also didn’t rescue any in vitro parameters). However, in light of our new immunoblot data, which does not show any significant differences in protein expression among the dynein mutants, we have changed our thinking. We now believe that the H3639P mutation simply compromises dynein folding, and this is the underlying reason for mistargeting and poor activity metrics. We have changed the language within the manuscript to reflect these new data and modified our conclusions accordingly.20) It would be more transparent to plot most of the data points either directly on to the bar charts or replace the bar charts with box and whisker plots.To improve transparency, we have added two new things to the figures: (1) as mentioned above, we have overlaid data points on plots in Figure 2, 3 and 4 that represent the mean values of each independent replicate; and, (2) we have included as supplementary material scatter plots displaying all data. Specifically, we added scatter plots for the following datasets:a) Velocity and displacement for spindle dynamics assay for haploid cells (new Figure 2—figure supplement 2A and B)b) Velocity and displacement for spindle dynamics assay for diploid cells (new Figure 3—figure supplement 1)c) Velocity and displacement for single molecule assays (new Figure 4—figure supplement 1A and B)d) Velocity and displacement for spindle dynamics assay for H3639P and related mutants (new Figure 6—figure supplement 1C and D)e) Velocity and displacement for single molecule assays for H3639P and related mutants (Figure 6—figure supplement 1H and I)f) Velocity and displacement for spindle dynamics assay for R1852C and related mutants (new Figure 7—figure supplement 2A and B)g) Velocity and displacement for single molecule assays for yeast (and human equivalent of) R1852C (R1962C) and related mutants (Figure 7—figure supplement 2C - E)21) The Abstract states there is no correlation between mutations and disease state, but then the authors find one. Should the first statement be "There is no known correlation [...]"?

Indeed. Per the reviewer’s suggestion, we changed the Abstract accordingly.

22) Figure 1: the authors should add the meaning of CC and DD in the legend of panel A and use the green color to indicate the bar for K1475Q (mutation in the linker domain) in panel B.Per the reviewer’s suggestion, we defined “CC”, “DD” (and even “MTBD” which we noted was lacking) in the Figure 1 legend. We thank the reviewer for catching this oversight on our part. We also added a distinct color to clearly differentiate the linker domain mutation (K1475Q) throughout all relevant figures. We thank the reviewer for the suggestion.23) The authors should try and remove some redundant wording (e.g. subsection “Single molecule motility assays reveal insight into dynein-intrinsic dysfunction “: the content of the parentheses in the first paragraph "e.g. Pac.1..." is redundant in light of the following two paragraphs) and modify some sentences (e.g. the kymograph in Figure 4—figure supplement 1 readily displays an increase in speed of K1475Q motors; however, it is difficult, if not impossible, to evince from this figure an increase in run length or in the fraction of active motors).

We would prefer to leave the parenthetical note in question as is. Although we discuss two of the molecules in that paragraph in more detail, the others are not mentioned. We think it best to use the short parenthetical note to allude to the complexities underlying dynein regulation in cells (and to include important citations).

We would also prefer to leave the note in the manuscript referring the reader to the example kymograph for K1475Q (in Figure 4—figure supplement 1C). Although we don’t expect the reader to evince fraction of active motors from this kymograph, we believe it nicely shows the increased run lengths and velocity parameters for this motor (note the large absence of shorter runs in this example with respect to wild-type).24) Likewise, the authors should consider being a little bit more cautious in some of their conclusions (e.g. it is almost impossible to rule out a contribution of adaptors and accessory factors to the phenotype observed in mammalian cells) or speculations (geometric constrains limiting the amount of motor complexes engaged in transport of a vesicle – an organelle is almost never perfectly round, which the exception of synaptic vesicles which are assembled after transport at synapses).We have softened the language in question concluding that altered in vitro dynein motility parameters are necessarily indicative of dynein-intrinsic defects. We have also softened our speculation that geometric constrains of a vesicle limit the number of motors that can engage with a microtubule. We thank the reviewer for the suggestions.25) Would it be possible to remove the term i.o. (insufficient observations) from some of the Figures (e.g. Figure 5) by increasing the number of observations or at least state the number of observed events in the figure legend?With regard to increasing the number of observations for those mutants that exhibit fewer events (i.e., neck transit events in Figure 2) or foci (Figure 5), we think it would be unfair to sample a greater number of cells for some mutants than we did for others. We did our best to keep all sample sizes roughly equal. Of course, this means that some data sets will be smaller than others, such as in the case of dynein foci observations (since some mutant foci are present at lower frequencies) or in the case of neck transit events (since H3639P has such a low degree of dynein activity). However, as suggested by the reviewer, we have now included information in the figure legend describing the number of observed events for these two examples.26) The readability of panels B and F of Figure 7 should be improved.We have increased font sizes in the panels in question to improve the readability of this figure. We thank the reviewer for the suggestion.27) The CDD calculation method should be included in the Material and methods section of the main text.We have now included our method of CDD calculation in the Materials and methods section. We thank the reviewer for the suggestion.28) The authors should consider including in the reference section some early references demonstrating the key role of cytoplasmic dynein in the nervous system of Metazoa (e.g. Saxton's lab, Drosophila; Fisher's lab, mouse), and some key reviews discussing the links between deficits in axonal transport and nervous system disorders, and the complexity of the clinical phenotypes caused by mutations in the DYNC1H1 gene.Per the reviewer’s suggestion, we have added references to important papers describing the role of dynein in neuronal transport, in its role in supporting neuronal health, and its implication in nervous system disorders. We apologize for this oversight, as the work in question laid the foundation for our understanding of dynein’s role in neuronal health and neurodegenerative diseases. We thank the reviewer for the suggestion.29) Please justify the decision of using one-way ANOVA OR unpaired t-test OR Z-scores for panels of the same figure (e.g. Figure 2). To the readers, these panels are indistinguishable in nature (they are derived from the same experimental dataset). This comment applies to the majority of main and supplemental figures.

Thanks to the reviewer’s question, we learned that we were using ANOVA incorrectly. This test assumes equal variances, which is inappropriate for the data on which we were applying this test. We have corrected this mistake, and applied the following three tests:

1) Unpaired t test with Welch’s correction (a.k.a., Welch’s t test). We used this to determine significance between normally distributed datasets (i.e., velocity).

2) Mann-Whitney test. We used this to determine significance between non-normally distributed datasets (i.e., displacement per event). Consistent with others, we find these datasets are exponentially distributed.

3) Z score calculation. We used Z scores to determine significance between proportions, as is accepted practice (e.g., number of dynein events per minute per cell, fraction of neck transits that were successful, etc).

We have added a brief explanation for these tests in the Materials and methods section (under the “Statistical analyses” header). We thank the reviewer for asking this question, which prompted us to discover and correct our mistake.

[Editors' note: further revisions were requested prior to acceptance, as described below.]

Thank you for resubmitting your work entitled "Molecular basis for dyneinopathies reveals insight into dynein regulation and dysfunction" for further consideration at eLife. Your revised article has been favorably evaluated by Anna Akhmanova as the Senior Editor, a Reviewing Editor, and two reviewers.The manuscript has been improved but there are some remaining issues that need to be addressed before acceptance, as outlined below:1) Address the following comments about toning down the claims of novelty.Some of the claims about the novelty of the finding that there is a correlation between disease type and strength of the mutations' effects on dynein activity are too strong, and need to be toned down.For example, the authors write in the Abstract "In addition to revealing molecular insight into dynein regulation, our data reveal an unexpected correlation between the degree of dynein dysfunction and disease type".The last part of the sentence is misleading as Hoang et al., 2017, reached a similar conclusion:"Interestingly, of the 16 human or mouse mutations that support production of the recombinant dynein complex, the 6 with the strongest effects on dynein motility are associated with MCD in humans (Figure 6). […] These findings raise the possibility that MCD and SMALED are caused by different degrees of inhibition of dynein motility, rather than inactivation of distinct dynein-related processes".Whilst the confirmation and extension of this correlation by Marzo et al. valuable, it would seem uncharitable not to mention that it had been reported before (and is therefore not unexpected to find this in the current study).The authors could, for example, write in the Abstract that "our data strengthen the evidence that different disease types arise from different degrees of dynein function", and must include an acknowledgement in the main text (e.g. in the Discussion) that Hoang et al. proposed this previously based on their observations.There is a similar statement in the last paragraph of the Introduction that also needs toning down.

We completely agree with this assessment, and apologize for this oversight on our part. We have made changes throughout our manuscript to include discussion describing these prior observations by Hoang et al. In particular, we have: (1) made the recommended change to language in the Abstract; (2) added language to the final paragraph of the Introduction alluding to these findings; and (3) added a sentence in the Discussion noting these prior observations by Hoang et al. We thank the reviewers for pointing out this oversight on our part.

2) It was not clear to us from the legends to the new Figure 5—figure supplement 2 what the two images for each blot represent. Are these different blots of the same extracts, or from two different extracts per experiment? This information should be included in the legend.

To clarify, in the prior iteration of Figure 5—figure supplement 2, each independent immunoblot was arranged into two rows of 9 bands, each representing one of the 17 mutants plus wild-type (i.e., the top row included lysates from wild-type and 7 mutants up to W612C, and the bottom row included mutants from K1475Q to H3639P). We have rearranged the blots to avoid this confusion such that each independent blot is in a single row. We apologize for the confusing layout of the blots, and hope the revised version of this figure is more clear.